# How does a partner's motor variability affect joint action?

**Simily Sabu[1]\*, Arianna Curioni[1], Cordula Vesper[2,3], Natalie Sebanz[1], Günther Knoblich[1]**

**1** Department of Cognitive Science, Central European University, Budapest, Hungary, **2** Department of Linguistics, Cognitive Science and Semiotics, Aarhus University, Aarhus, Denmark, **3** Interacting Minds Centre, Aarhus University, Aarhus, Denmark

\* sabu_simily@phd.ceu.edu

**Data Availability Statement:** All relevant data are within the paper and its Supporting Information files.

**Funding:** This work was supported by the European Research Council under the European

## Abstract

Motor learning studies demonstrate that an individual's natural motor variability predicts the rate at which she learns a motor task. Individuals exhibiting higher variability learn motor tasks faster, presumably because variability fosters exploration of a wider space of motor parameters. However, it is unclear how individuals regulate variability while learning a motor task together with a partner who perturbs their movements. In the current study, we investigated whether and how variability affects performance and learning in such joint actions. Participants learned to jointly perform a sequence of movements with a confederate who was either highly variable or less variable in her movements. A haptic coupling between the actors led to translation of partner's movement variability into a force perturbation. We tested how the variability and predictability of force perturbations coming from a partner foster or hamper individual and joint performance. In experiment 1, the confederate produced more or less variable range of force perturbations that occurred in an unpredictable order. In experiment 2, the confederate produced more or less variable force perturbations in a predictable order. In experiment 3, the confederate produced more or less variable force perturbations in which the magnitude of force delivered was predictable whereas the direction of the force was unpredictable. We analysed individual performance, measured as movement accuracy and joint performance, measured as interpersonal asynchrony. Results indicated that in all three experiments, participants successfully regulated the variability of their own movements. However, individual performance was worse when partner produced highly variable force perturbations in an unpredictable order. Interestingly, predictability of force perturbations offset the detrimental effects of variability on individual performance. Furthermore, participants in the high variability condition achieved higher flexibility and resilience for a wide range of force perturbations, when the partner produced predictable movements. Participants improved their joint performance with a highly variable partner only when the partner produced partially predictable movements. Our results indicate that individuals involved in a joint action selectively rely on either their own or their partner's variability (or both) for benefitting individual and joint action performance, depending on the predictability of the partner' movements.

Union's Seventh Framework Program (FP7/2007-2013) / ERC grant agreement n˚ 609819, SOMICS, and by ERC grant agreement n˚ 616072, JAXPERTISE.

**Competing interests:** The authors have declared that no competing interests exist.

## Introduction

We engage in interactions with other people very often in our daily life. One type of such interactions consists of one person learning a motor skill with another person while their bodies are coupled through some means. Examples include learning to dance tango with a partner, a parent holding a child to help her walk, learning group sports like crew rowing, or a therapist moving a patient during motor rehabilitation training. Such inter-personal coupling implies that each actor's movement will have a direct impact on the partner's movements [1].

Generally, in such joint actions, individuals adopt various coordination strategies to minimize error and improve joint performance. Enhancing predictability of one's own movement is one such coordination strategy [2, 3]. Vesper and colleagues [2] demonstrated that when minimal perceptual information is available to coordinate the timing of discrete action outcomes, individuals make themselves more predictable to the co-actor by reducing the temporal variability of their actions. Similar results were obtained for more complex temporally extended joint actions [3].

Evidence for strategical variability reduction could be observed across a variety of joint action tasks [4–9]. Studies involving joint grasping have shown that when a dyadic interaction involves a leader-follower dynamic between the actors, the leader tends to reduce the variability of her movements as a signalling strategy to enhance predictability of her movements for the follower [6]. During joint force-production tasks, it was observed that participants reduce the variability of their movement duration and force production, compared to when they act alone [7, 8]. A study by Vesper and colleagues [9] showed that when dyads performed a movement task that required them to synchronously arrive at a target from separate starting locations, participants held their movement duration constant to facilitate coordination in cases when there was only auditory feedback available about a partner's movements. In sum, evidence from a wide range of joint action studies confirms that in coordination contexts variability in movements is detrimental for joint performances.

In contrast to the joint action literature, it was recently claimed that variability can facilitate individual motor learning. Motor variability need not necessarily be a noise in the motor system which should be minimised to achieve movement perfection, rather it could be exploited by individuals, thus, becoming a key factor enabling and facilitating motor learning and skill acquisition [10–17]. In support of this claim Wu et al., [10] showed that the temporal structure of motor variability naturally produced by individuals at an early phase of learning can predict the rate at which they will learn a motor task. It was shown that the rate of learning was higher in individuals who exhibited higher task-relevant variability during the early phase of learning. The study reveals an adaptive function of motor variability, as it supports a wider action exploration of motor commands: it may provide individuals with a repertoire of actions from which they can select to achieve a successful outcome in current and future movements. This adaptive role of variability has been observed in different types of learning regimes, such as reward-based learning and error-based learning [10, 14].

However, it needs to be noted that if variability of the motor outcome is random, it will not benefit learning, especially in types of learning that requires reduction of variability for achieving movement perfection. Ranganathan and colleagues [18, 19] propose that even though variability can enhance action exploration in certain types of learning, a higher magnitude of variability may adversely affect learning by leading to poor retention of learned solutions. Also, the learning will be adversely affected if the mechanism involved is use-dependant, i.e., if the learning requires one to produce subsequent movements similar to the previous ones or if it requires coordination pattern stability, as practising unstable movement patterns, can lead to poor learning. Similarly, Barbado and colleagues [14] showed that the magnitude of variability

in error-based learning only negatively influences the learning rate, while the structure of individuals' variability or the systematic variation in their movements enhanced their ability to detect motor error, which ultimately led to faster learning. Thus, it appears that it is the pattern of unfolding of variability, i.e., the *structure of variability*, that determines how individuals learn to refine their motor output over time, rather than the magnitude of variability itself.

The improved rate of learning in the presence of a structured high motor variability could be explained by how motor events unfold during an action exploration phase. During action exploration, individuals vary their motor parameters in search of action solutions that may yield the best outcome. These aids wider sampling of action possibilities that could be exploited in future performance of such actions. There is evidence suggesting that the nervous system actively produces variability in motor output as a means of searching for actions that ensure greater success during motor learning [11, 20]. Exposure to variable motor parameters in a motor learning task would allow one to perfect a more flexible internal model of the task, which in turn may allow one to generalize or adapt the learned skill to varied task demands [21]. Thus, action exploration may lead to the generation of a flexible internal model that accommodate a wider range of movement solutions.

Taken together, previous research indicates that variability plays differential roles when performing joint and individual actions: when performing joint actions, reducing the variability of one's movements seems to optimize the joint performance; in individual motor learning, high variability seems to be beneficial for learning, at least under some conditions. The present study asked whether higher variability may be beneficial during *joint motor learning* where an individual learns to perform a motor task together with a partner. This would be expected especially when a joint action involves tight physical interaction between two partners where one partner's movements directly affect the other partner's movements, potentially perturbing the other's movement trajectory. Since the systematic variation or the structure of variability is a significant factor determining the functional role of variability in learning, and since predictability of partner's movements are crucial for successful joint actions, the partner's variability was delivered at different degrees of predictability or structure. It is not known whether in this type of joint action, actors will reduce the variability of their movements to maximize the efficiency of joint action or whether they will exploit the variability of their partner's movement to improve their own learning performance or both.

## The present study

In the present study, we aimed to investigate whether individuals can utilize their partner's variability for motor learning when repeatedly performing a motor task together. Specifically, we investigated whether individuals could benefit from a partner's motor variability in the context of performing a joint task, depending on the predictability of the range of perturbations. If a partner exerts a highly variable range of perturbations on an individual's movements, one could predict that a partner's variability helps an individual engaged in a joint action to explore a larger space of movement parameters [10]. This could allow her to explore a broader range of movement possibilities that can be performed to contribute to a joint goal, as well as the individual goal. If the partner exerts a small range of perturbations, the individual will only explore a narrow range of possible movement. She may therefore become less adaptive compared to someone who learned with a highly variable partner.

Because joint action requires the actor's movements to be predictable, one should expect beneficial effects of a partner's variability only in joint contexts when the individual can predict which movement the partner is going to perform and what perturbation from the partner can be expected. Thus, when a partner exerts high variability on an individual's movements in a

predictable manner, this should benefit individual learning and individual contributions to the joint task. Alternatively, if the partner's high variability reduces the predictability of her upcoming movement, then the higher variability should interfere with the individual's learning and contribution to the joint task. Also, even when the partner exerts variability in a predictable manner, individuals still would need to adopt some motor strategies to account for the differential effect of variability in individual and joint action scenarios.

To test these predictions, we developed a new joint sequence learning task, in which participants learned to perform a sequence of aiming actions while being haptically coupled to one another. We manipulated the variability and predictability of a partner's perturbations exerted on participants' movements. This allowed us to investigate the differential contribution of variability (range of perturbations experienced) and predictability (order of perturbations delivered) of perturbations to the individual's learning performance and the actors' joint performance during the joint action. Experiment 1 investigated whether individuals can benefit from high variability of their partner's movements while the partner's movement is completely unpredictable. Experiment 2 looked at whether predictability of a partner's movements enhances individuals' ability to utilize partners' high variability to improve or stabilize their own and joint performance. Experiment 3 asked whether partial predictability of a partner's movements is sufficient to obtain benefits from partner's high variability for the individual and joint performance. The study treats spatial accuracy of individuals as a measure of their individual performance and the inter-personal asynchronies between the landing times as a measure of their joint performance.

## Experiment 1

In Experiment 1, we investigated whether high variability of partner's movements can aid performance in individuals, while a partner's upcoming movements are completely unpredictable. Participants performed a joint task that consisted in producing a sequence of joint force configurations with a partner (see Fig 1A). The partner was a confederate producing either highly or less variable force perturbations on the participants' movements throughout a sequence. The properties of force perturbations coming from the partner were completely unpredictable because the partner moved to a different sequence of locations in each trial. Following Wu and colleagues' [10] account one could expect a beneficial role of high variability in individual motor learning in joint motor learning scenarios, even if a partner's movement sequence is not predictable. Individuals performing the task while experiencing a high variability in one's movements may still benefit from enhanced opportunities for action exploration. Accordingly, performing with a highly variable partner may lead to enhanced performance through better learning than performing with a partner producing low variability (a lower range of force perturbations on the participant's movement). However, because predictability is a crucial precondition for successful joint action coordination [2, 3], it is also possible that being able to predict a partner's next movement while acting together in a joint context is a precondition for any potential benefit arising from an enhanced range of force perturbations experienced. Also, it has been shown that it is the structure of variability and not its magnitude that potentially facilitates learning [14]. If this is so, individuals' performance may suffer from performing joint actions with a highly variable partner whose upcoming actions cannot be predicted.

## Materials and methods

### Ethics statement

The study was approved by the Hungarian ethics committee (United Ethical Review Committee for Research in Psychology- EPKEB). All participants gave informed consent in written

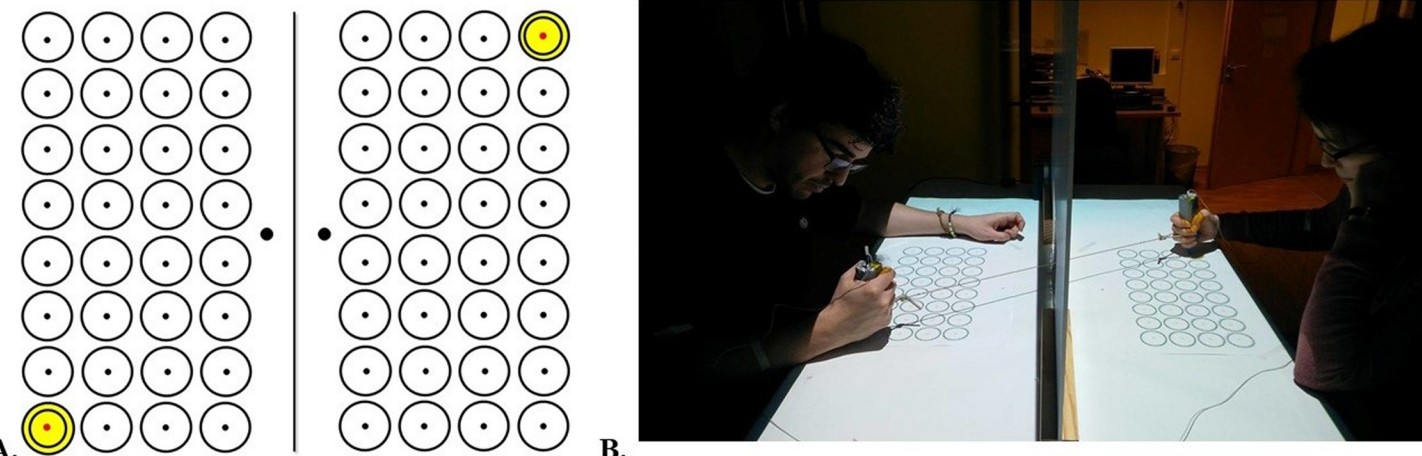

**Fig 1. Experimental apparatus and stimuli.** (A) 4 x 8 arrays of possible target locations for the two actors; targets were projected as yellow concentric circles with a red centre. The diameter of each of the outer circle was 4.5 cm and that of the red center was 0.5cm. A black dot above each target array marked the starting position for the movement. Two simultaneously cued target locations for the confederate and the participant constitute a *configuration*. (B) Figure shows actors haptically coupled by an elastic band, while performing in the joint action phase in which each actor had to hit the red center of their own target circles, synchronising with their partner, using the handles provided to them. The individuals from the photograph has given written informed consent (as outlined in PLOS consent form) to publish this photograph.

form prior to the experiment. The individuals from the photographs (Fig 1B) gave their written informed consent (as outlined in PLOS consent form) to publish their photographs.

## Participants

Forty people participated in the study (23 Females, Mean age = 27.5 years, SD age = 1.98 years). 4 participants were excluded from the analysis as they failed to complete the task. Participants were recruited through the SONA online participation system. All participants were right-handed and reported to have normal or corrected-to-normal vision. They received monetary compensation for their participation. Two lab assistants volunteered as confederates for the experiment.

## Apparatus and stimuli

Two actors, a participant and a confederate sat on either side of the experimental table facing each other. Actors were haptically coupled using an elastic rubber band. Thirty-two circles were arranged on the table in a 4x8 array in front of each actor and defined possible target locations for participants' movements (see Fig 1A). Targets were projected using an Epson EH-TW490 Lumen projector attached to the ceiling and cued as yellow concentric circles with a red centre.

A Polhemus G4 electro-magnetic motion capture system (40 Hercules Drive, Colchester, Vermont) was used to track the movements of the two actors. Actors were given a handle with a micro-motion sensor (1.8 mm) inserted inside to perform the task (see Fig 1B). The actor's movements were tracked at a frequency of 120 Hz. An occluder was placed between the actors to prevent them from seeing each others' movements. An opening in the bottom of the occluder allowed haptic coupling between the two actors. A stretchable rubber band was used to create the haptic coupling between the two actors. The experiment was run on a Dell Precision computer. MATLAB (R2015a) and RStudio Team (2020) was used for running the experimental script and for performing the data analyses. A force measuring gauge, Sauter FK250 Digitális Erőmérő, was used to measure the forces produced by the elastic band at each target configuration, in order to create the sequences necessary for the experiment.

## Procedure

**The joint task.**　A participant and a confederate performed a *joint aiming task* together. Both actors performed their actions on the target array on their side of the table (see Fig 1A). Every sequence was composed of 8 target locations for both participant and confederate (we will refer to *configuration* to indicate the ensemble of participant's and confederate's targets). For every sequence, each of the eight-target location was cued for 1000 ms simultaneously for the participant and the confederate. Participants were instructed to hit the center of the target location on their side as accurately as possible within the cueing duration and to land as synchronously as possible with their partner at the target location.

**Variability manipulation.**　The experiment was designed to have a between-subject manipulation with two experimental groups, **High Variability (HV)** and **Low Variability (LV) group**. The target sequences performed by participants in both groups were the same, while the confederate's sequences were manipulated to induce high or low variability. Variability was determined by the standard deviation of the inter-personal distances of within each sequence: it was higher (SD = 9.55cm) for high variability sequences and lower (SD = 2.22 cm) for low variability sequences. The haptic coupling between the two actors ensured that movements of one person impacted the other. Thus, variable or constant inter-personal distances across configurations within a sequence resulted in corresponding high variable or low variable force perturbations on the participant's movements (see Fig 2A and 2B). However, the average inter-personal distances, within every sequence, in both groups were comparable (Mean Inter-personal Distance for HV = 50.22 cm and LV = 49.77 cm).

Predictability of a partner's movements was disrupted by breaking any regularity of the vector properties of force perturbations is coming from the partner (direction and magnitude of

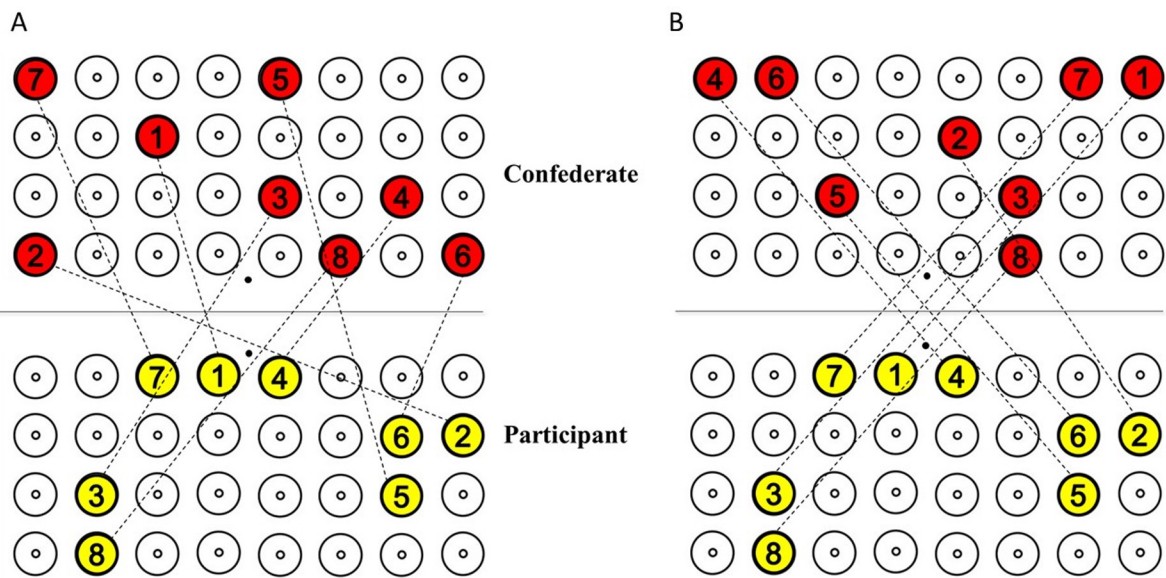

**Fig 2. Example of a HV and LV sequence.** Red circles denote confederate's targets and yellow circles denote participant's targets. Black dashed lines represent various target configurations within the sequence. (A) High variability sequence with varying inter-personal distances across all eight configurations in a sequence. (B) Low variability sequence with similar inter-personal distance across all eight configurations. Average inter-personal distances of the configurations for the two groups are the same. Participants in the HV and LV group received the same targets. Confederate's target locations were manipulated to produce sequences with different degrees of variability and predictability.

the forces). At each repetition of participant's sequence, the confederate was given a different sequence to perform both in the HV and LV group. In each block, participants repeated a single sequence 10 times, but the confederate performed 10 different movement sequences (see Fig 3). Thus, the force perturbations exerted by the confederate across every repetition of a sequence were unpredictable. The order of the blocks was counter-balanced across participants. Short breaks were taken in between blocks.

**Experiment timeline.** The experiment consisted of 8 blocks in which each block corresponded to a specific sequence of target configurations. Each block consisted of three phases as following: *Demonstration phase*: participants observed the upcoming sequence of their individual targets in the sequence. Each target was highlighted for a duration of 1000 ms followed by the next target. *Individual action phase*: participants individually performed the sequence they needed to perform during the joint action, without experiencing any force perturbation. This phase was introduced only to familiarize the participants with the aiming task they need to perform. Participants started a trial from the starting position, and then went on hitting the eight target circles as they lit up one after the other. Participants were instructed to hit each target within its 1000 ms cueing duration. After the presentation of the eighth target, the whole array of target circles turned black. During this interval, the actors had to get back to their start position (See Fig 4). Participants repeated the sequence 5 times.

*Joint phase*: Participant and confederate performed the task together. In this phase, the actors were haptically coupled. They were instructed to hit the centre of the cued targets as accurately as possible, and to be temporally synchronized with their partner, i.e., hit their respective targets at the same time. Within a single block, participants repeated the corresponding sequence 10 times, while confederate performed the sequences according to the variability condition, as detailed in the previous section. The 32 potential target positions for each actor were calibrated as reference coordinates for every duo before the start of the experiment. The overall duration of the experiment was about 35 minutes.

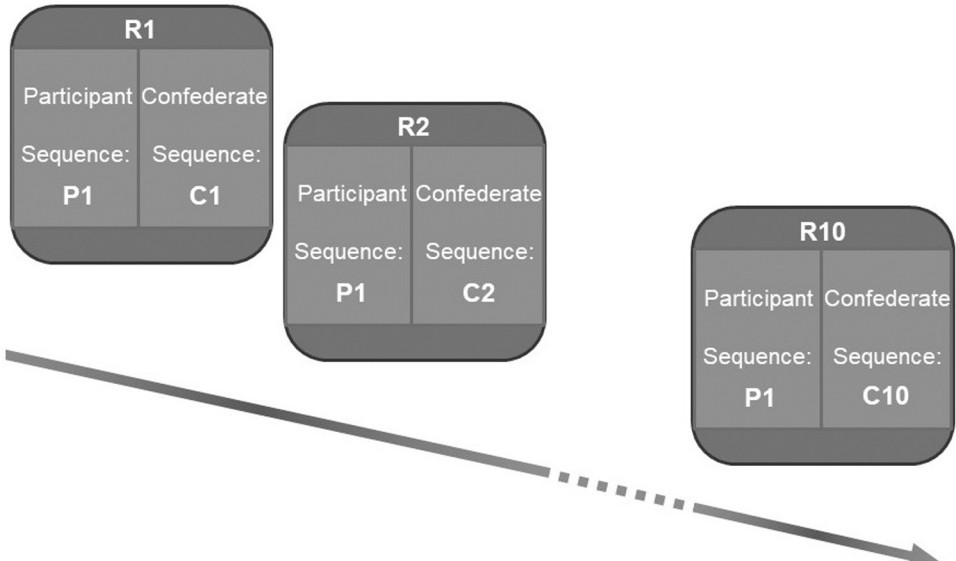

**Fig 3. Manipulation of variability and predictability in Experiment 1.** Only a single block is shown in the figure. *Variability* in both groups are manipulated by varying the inter-personal target distances. Predictability was manipulated by making the participant repeat the same movement sequence (P1) across repetitions R1-R10, while the confederate performs 10 different sequences (C1-C10) across R1-R10, to produce *completely unpredictable* movements.

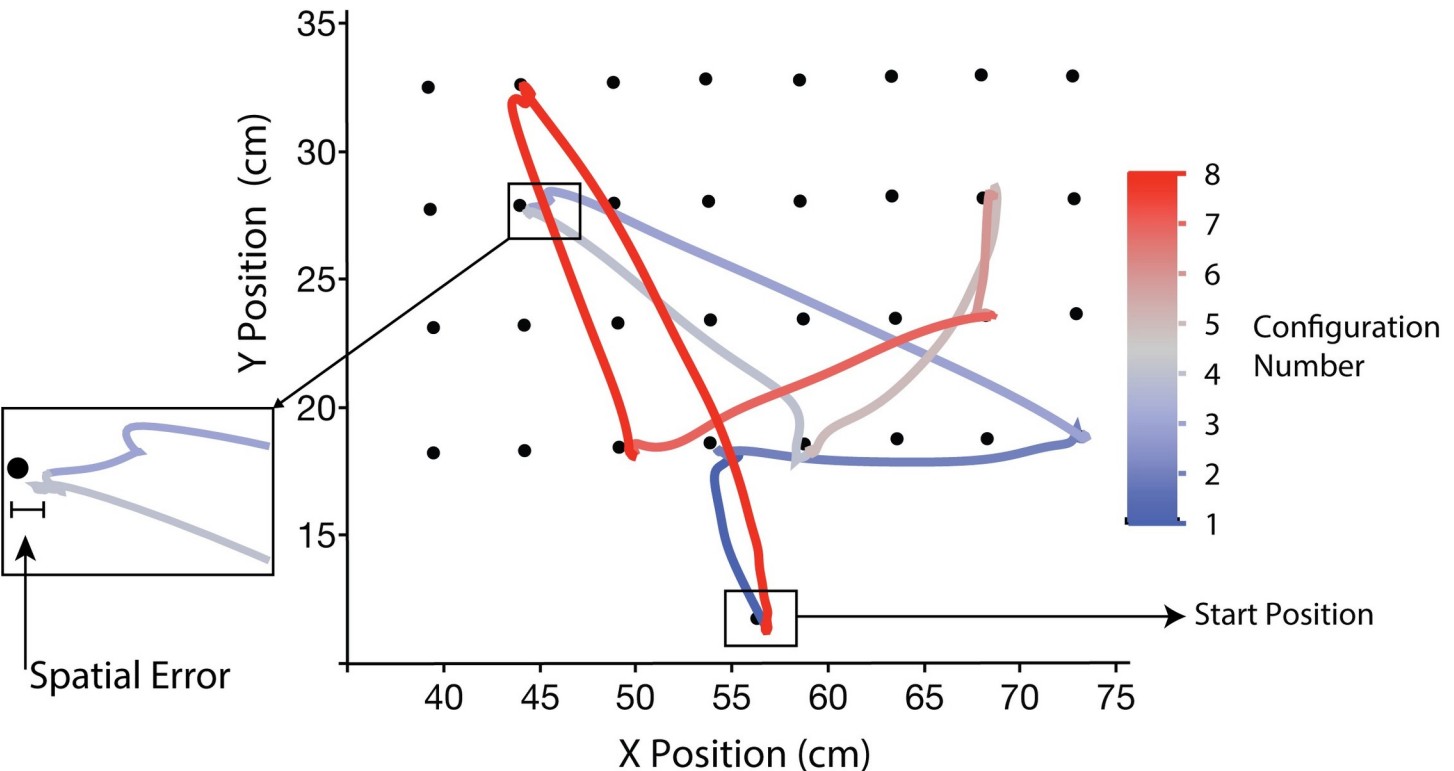

**Fig 4. Example trajectories of participant's movements in a joint-action phase.** The figure shows the kinematic data of a participant across all eight targets of a movement sequence. Spatial error marks the distance between the target position that participants were supposed to hit and where they hit at the end of each trajectory which is referred to as landing position.

**Data processing and analysis.** We segmented the participant's and confederates' kinematic data by retrieving the landing positions, which are the final position of the handle after an actor hit the target (target-to-target movement). The landing positions were identified as the 2D coordinates of the actor's movements when their movement velocity (m/s) was minimal (~0 cm/sec) for each configuration, using an automatized algorithm in MATLAB. The landing positions were used to measure the spatial error produced by the actors and also the variability of participant's movements on the horizontal and vertical dimension.

Participants' ***movement variability on horizontal and vertical dimension (cm)***, normalized over the distance covered in each trajectory, were computed as a measure of the variations that occurred in their motor performance. The variability was calculated as the average deviations of the spatial coordinates on the two spatial components of their movements separately, namely the horizontal and the vertical dimension, divided by the distance covered in each trajectory. The measure was subjected to a mixed 2x2x10 ANOVA with Group (High Variability (HV) and Low Variability (LV) group) as between-subject factor and the Axis-dimension (horizontal and vertical) and Repetition (R1-R10) as the within-subject factors. Greenhouse-Geisser corrected values were used for factors that violated Mauchly's sphericity test for every analysis.

As a measure of individual performance, the ***spatial error (cm)*** of the actor's hits were obtained by calculating the Euclidean distance between the landing position and the actual target position. Trials in which actors moved to positions 5 cm or more away outside the target array were considered as outliers and removed from the analysis. 0.13% trials were removed from HV group and 0.27% trials were removed from LV group through outlier removal. The reduction of spatial error over repetitions were analysed to see how the actors learned to

predict the force configurations. The mean spatial error across sequences in all 8 blocks was calculated for each of the 10 repetitions of the sequences and was subjected to a mixed 2x10 ANOVA with Group (HV and LV) as the between-subject factor and Repetitions (R1-R10) as the within-subject factor. The spatial error of participants was also correlated with the magnitude of force perturbations experienced by the participants from the partner, to understand how the participant's motor performance changes with different forces. A repeated measures correlation (rmcorr) was computed to assess the correlation between the spatial error across different magnitude of force perturbation, in the first (R1) and last repetition (R10) separately. Outliers were removed from the correlation analysis using (mean +/- 2.5 SD) rule. 3.03% trials and 3.5% trials from the repetitions R1 and R10 respectively of HV group and 0.61% trials from R1 of LV group was removed through outlier removal.

To assess the joint performance, ***Absolute Asynchronies (sec)*** between the actors were computed as a measure of inter-personal temporal coordination. The absolute asynchronies were derived from the raw asynchronies that were calculated as the difference between the time point of the participant's landing time and the confederate's landing time. The asynchronies were subjected to a mixed 2x10 ANOVA with Group (HV and LV) as the between-subject factor and Repetitions (R1-R10) as the within-subject factor. Greenhouse-Geisser corrected values were used for factors that violated Mauchly's sphericity test.

All significant main effects and interactions were further analysed by applying Bonferroni correction for multiple comparisons.

## Results

### Movement variability on horizontal and vertical dimension

The mixed ANOVA revealed a main effect of Repetition with Greenhouse-Geisser correction ($\varepsilon = 0.721$), indicating that participants in both groups reduced their variability over time ($F_{(9, 306)} = 7.926$, $p < 0.0001$, $\eta^2 = 0.189$, see Fig 5A). Post-hoc analyses revealed that variability at first repetition, R1 (mean = 0.120, SE = 0.007) was significantly higher compared to other repetitions (all $ps < 0.005$). The main effect of axis-dimension was also significant ($F_{(1,34)} = 22.885$, $p < 0.0001$, $\eta^2 = 0.402$) with the variability on the horizontal dimension being larger (mean = 0.091, SE = 0.004) than the variability on the vertical dimension (mean = 0.074, SE = 0.004). The analysis also showed a significant main effect of group ($F_{(1,34)} = 11.667$, $p = 0.002$, $\eta^2 = 0.255$), indicating that the HV group participants had a higher movement variability (mean = 0.095, SE = 0.005) compared to the LV group (mean = 0.069, SE = 0.005). The interaction between repetition and group ($F_{(9,306)} = 2.443$, $p = 0.011$, $\eta^2 = 0.067$) was significant with Greenhouse-Geisser correction ($\varepsilon = 0.509$), with the largest mean difference driven by the higher variability of HV group participants at R1 (mean = 0.148, SE = 0.011) compared to LV group (mean = 0.093, SE = 0.011). This was confirmed by post-hoc analysis ($p = 0.001$). This indicates a faster reduction of variability by HV group compared to LV group. All other interactions were non-significant ($ps > 0.06$; see S1 File).

### Spatial error

The ANOVA with a Greenhouse-Geisser correction ($\varepsilon = 0.583$) revealed a main effect of the repetitions, indicating that participants in both groups learned to reduce spatial error over time ($F_{(9,306)} = 27.173$, $p < 0.0001$, $\eta^2 = 0.444$). Post-hoc analysis revealed that the spatial error at R1 (mean = 1.240, SE = 0.055) was higher compared to all other repetitions, R2-R10 (all *ps* < 0.001). The analysis also showed a significant main effect of group, with the HV group (Mean = 1.03, SE = 0.05) having significantly larger errors than the LV group (Mean = 0.857, SE = 0.05), ($F_{(1, 34)} = 6.089$, $p = 0.019$, $\eta^2 = 0.152$; See Fig 5B). The interaction between the two

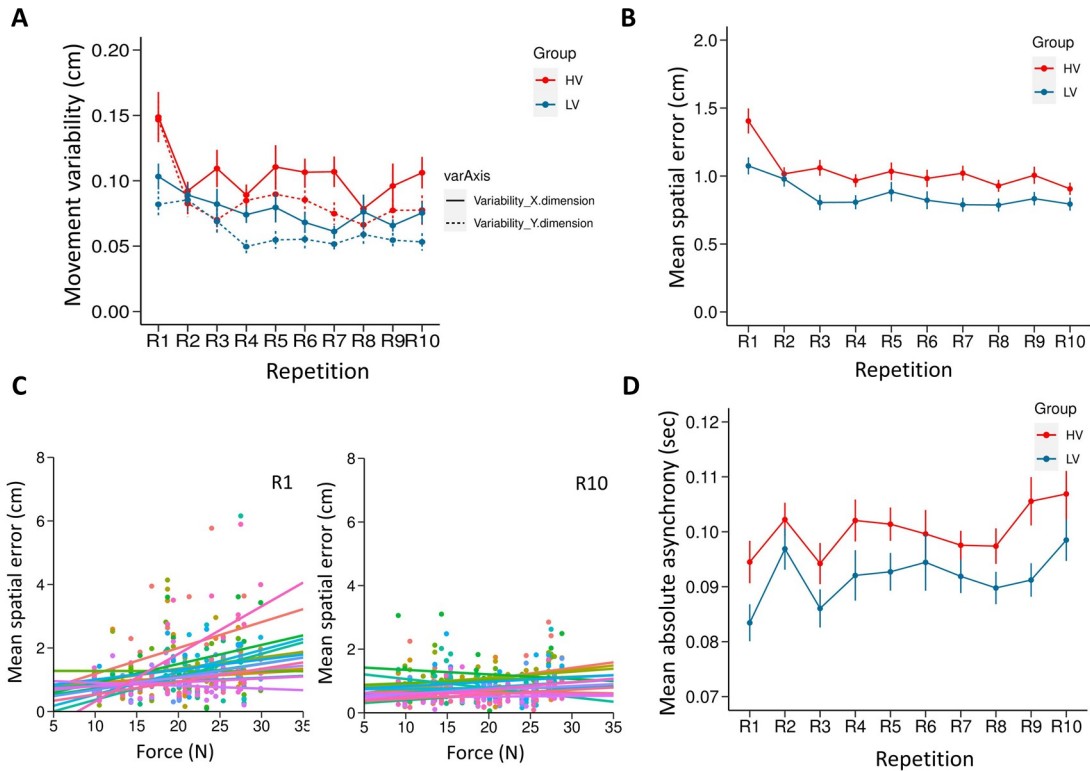

**Fig 5. Results of Experiment 1.** (A) Normalized variability of participant's movement in the horizontal and vertical dimension. (B) Spatial error as a measure of individual performance. The spatial error at each repetition is an average of all 8 configurations from each of the 8 blocks. (C) Repeated measures correlation analysis between the force perturbation experienced and the spatial error of the participants in the HV group, for R1 and R10 separately. Each participant's data and corresponding rmcorr fit lines are shown in different colors. (D) Absolute Asynchronies as a measure of joint performance. All error bars display standard error of mean.

factors was also significant (F (9, 306) = 3.503, p = 0.0004, $\eta^2$ = 0.093). Initial reduction in error from the first to the second repetition was larger in the HV group (mean = 1.405, SE = 0.078) than in the LV group (mean = 1.075, SE = 0.078), which was confirmed by the post-hoc analyses (p = 0.005). The analysis on confederate's spatial accuracy could be found in the (S2 File).

A repeated measures correlation (rmcorr) was computed to determine the relationship between force perturbation and the spatial error in the HV group participants. The correlation was done on the first (R1) and last repetitions (R10) separately. The analysis revealed a significant correlation between the force perturbation experienced and the spatial error at R1 ($r_{rm}(364)$ = 0.290, p < 0.001, 95% CI = 0.1935913 0.3819815) and as well as at R10 ($r_{rm}(414)$ = 0.126, p = 0.009, 95% CI = 0.03092686 0.2206055). A scatter plot (see Fig 5C) summarizes the results. The results show that participants made more errors when they experience larger force perturbations. As the LV group experienced a significantly lower range of varying forces leading to comparatively smaller number of force magnitudes, the correlation on the LV group is not of direct interest in the main discussion. Hence, the analysis on LV group in the current and following experiments are included in the (see, S3 File).

## Absolute asynchronies

The ANOVA with a Greenhouse-Geisser correction (ε = 0.729) on the absolute asynchronies between the two actors revealed a main effect of Repetition (F (9, 306) = 5.959, p<0.0001, $\eta^2$ =

0.149, see Fig 5D). The post-hoc revealed that the effect was mediated mainly by R1 (mean = 0.089, SE = 0.003) being significantly lower to R2 (mean = 0.100, SE = 0.002), R9 (mean = 0.098, SE = 0.003) and R10 (mean = 0.103, SE = 0.003) (all $ps < 0.01$). The main effect of mean asynchronies between HV group (mean = 0.100, SE = 0.003) and LV group (mean = 0.092, SE = 0.003) showed a trend towards significance (F (1,34) = 4.117, p = 0.0503, $\eta^2$ = 0.108). The interaction between the two factors was not significant (F (9,306) = 0.701, p = 0.708, $\eta^2$ = 0.020).

## Discussion

In the current study we investigated whether high variability of a partner's movements can aid performance in individuals while learning a motor task together. In our task successful individual performance or learning implies improvement in the ability to compensate force perturbations from a partner over time, by modulating one's own movements. Our results indicate that having a partner producing highly variable movement in an unpredictable order negatively affects individual performance compared to having a partner producing movements that are less variable. The correlation analysis revealed that participants in the high variability group had difficulty in adapting to larger force perturbations. There were no indications of a beneficial effect of variability in joint action contexts where a partner performs movements in an unpredictable order. One possible explanation for this result is that in the high variability condition, there was hardly any opportunity to improve predictions of a partner's behavior as a different force perturbation occurred at every trial. The HV group also suffered detrimental effects of partner's variability in their joint performance, compared to LV group, which can be observed from their difference in asynchrony measures. However, it is important to note that even though participants in the high variability group were consistently less accurate in their performance, they reduced their spatial error over time at a rate comparable to participants in the low variability group. The learning could have been achieved by successful reduction of individual's own movement variability over time, which is observed from the variability analysis. It should also be noted that participants were generally more variable on their horizontal dimension compared to the vertical dimension, owing to the possibility of participants exploring their motor parameters on the dimension that was less constrained by the haptic coupling. This indicates that a highly variable but unpredictable partner does not preclude learning during joint action, however, does not contribute to learning either. Taken together, results of Experiment 1 shows that having a partner who exerts high variability on one's movements in an unpredictable manner does not benefit individual or joint performance.

## Experiment 2

In this experiment, we investigated whether predictability of a partner's movement sequence may be a necessary precondition to observe beneficial effects of variability in a joint action scenario. Variability was manipulated as in the previous experiment with the partner producing small or large range of force perturbations. However, in Experiment 2, the partner produced predictable force perturbations in both the low and high variability conditions because he repeated the same movement sequence. If predictability of a partner's movements is a necessary precondition for individuals to exploit a partner's high variability, one should expect that participants in the high variability group should exhibit better individual performance in Experiment 2, as higher variability allows individuals to explore their action space and learn multiple action possibilities while the predictability of a partner's moves will aid the joint action. Alternatively, if predictability does not serve any benefits to utilizing partner's

variability, one should expect that partner's high variability cannot be beneficial in a joint context and hence we should see the same pattern of results as in Experiment 1.

## Materials and methods

### Participants

Forty people participated in the study (25 Females, Mean Age = 25.42 years, SD age = 3.1 years). All participants were right-handed and reported to have normal or corrected-to-normal vision. A confederate was hired to participate throughout the study for all 40 participants. They received monetary compensation for their participation.

### Ethics statement

The study was approved by the Hungarian ethics committee (United Ethical Review Committee for Research in Psychology- EPKEB). All participants gave informed consent in written form prior to the experiment.

### Procedure

**The joint task.** The joint task was the same as in Experiment 1, in which both the participant and the confederate performed the joint aiming task together while being haptically coupled.

**Variability manipulation.** Variability of the confederate's movements were manipulated as in Experiment 1. The standard deviation of the inter-personal distances within every sequence was manipulated to induce the difference in confederate's variability across groups. The standard deviations for high variability sequences was 11.85 cm and for low variability sequences 0.46 cm. However, the average inter-personal distances, within every sequence, in both groups, were maintained the same as in Experiment 1 (Mean Inter-personal Distance for HV = 52.13 cm and LV = 51.88 cm). Differently from Experiment 1, in the current study, the movement sequence confederates performed was predictable (see Fig 6). This also made upcoming force perturbations more predictable. The confederate repeated the same movement sequence within a block of trials. Thus, in a single block, both the participant and the confederate received a single sequence (8 target configurations) at every repetition.

**Experiment timeline.** As in Experiment 1, current experiment consisted of 8 blocks with each block corresponding to a specific sequence of target configurations. Each block involved three phases as following: *Demonstration phase-* participants observe the sequence of targets corresponding to the current block. Each target was highlighted for a duration of 1000 ms followed by the next target. *Individual action phase*—participants practised the sequence they needed to perform during the joint action repeatedly for 5 times, without experiencing any force perturbation. *Joint phase-*Participant and confederate performed the task together. Actors were haptically coupled and were instructed to hit the centre of the cued targets as accurately as possible and to be temporally synchronized with their partner, i.e., hit their respective targets at the same time. Actors were supposed to hit the targets within the 1000 ms cueing duration provided to them. Within a single block, participants repeated the corresponding sequence 5 times, differently from the 10 repetitions in Experiment 1. The number of repetitions were reduced in the current experiment to ensure that there were no performance decrements across repetitions due to fatigue. The confederate performed the sequences according to the variability condition, as detailed in the previous section. The 32 potential target positions for each actor were calibrated as reference coordinates for every duo before the start of the experiment. The overall duration of the experiment was about 25 minutes.

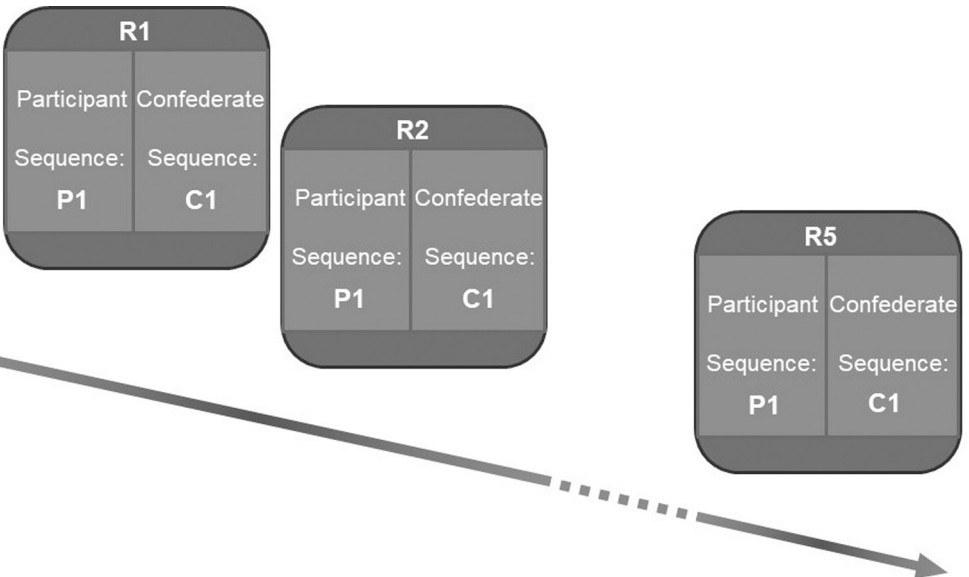

**Fig 6. Manipulation of variability and predictability in Experiment 2.** Only a single t block is shown in the figure. Variability in both groups are manipulated by varying the inter-personal target distances. Predictability was manipulated by making the participant repeat the same movement sequence (P1) across repetitions R1-R5, while the confederate also repeats her sequence (C1) across R1-R5, to produce completely predictable movements.

**Data processing and analysis.** Participant's and confederates' kinematic data was segmented using their landing positions, as in Experiment 1. The landing positions were used to measure the spatial error produced by the actors and the variability of participant's movements on the horizontal and vertical dimension. 0.17% trials from HV group and 0.02% trials from LV group was removed as outliers.

Participants' ***movement variability on horizontal and vertical dimension (cm)*** were computed as a measure of the variations that occurred in their motor performance. The variability was calculated as the average deviations of the spatial coordinates on the two spatial components of their movements separately, namely the horizontal and the vertical components, and normalized to the distance travelled in each trajectory. The measure was subjected to a mixed 2x2x5 ANOVA with Group (HV and LV) as between-subject factor and the Axis-dimension (horizontal and vertical) and Repetition (R1-R5) as the within-subject factors. Greenhouse-Geisser corrected values were used for factors that violated Mauchly's sphericity test for all analyses.

***Spatial error (cm)*** of the actor's performance was subjected to a mixed 2x5 ANOVA with Group (High Variability (HV) and Low Variability (LV) group) as the between-subject factor and Repetitions (R1-R5) as the within-subject factor. The spatial error of participants was subjected to a repeated measures correlation (rmcorr) along with the magnitude of force perturbations experienced by the participants from the partner, to understand how the participant's motor performance changes with different forces. The rmcorr was performed on first (R1) and last repetition (R5) separately, as in Experiment 1 (outliers were removed using (mean +/- 2.5 SD) rule). 4.38% trials and 2.50% trials from repetitions R1 and R5 respectively of the HV group was removed through outlier removal.

***Absolute asynchronies (sec)*** between the actors were retrieved as a measure of inter-personal temporal coordination by computing the difference of participant's and confederate's landing times and was subjected to a 2x5 ANOVA as for the spatial error measure.

All significant main effects and interactions were further analysed by applying Bonferroni correction for multiple comparisons.

## Results

### Movement variability on horizontal and vertical dimension

The mixed ANOVA revealed a main effect of Repetition with Greenhouse-Geisser correction ($\varepsilon = 0.705$), indicating that participants in both groups reduced their variability over time ($F_{(4,152)} = 13.384$, $p < 0.0001$, $\eta^2 = 0.260$, see Fig 7A). Post-hoc analysis revealed that the variability at first repetition, R1 (mean = 0.092, SE = 0.008) was significantly higher compared to other repetitions (all $ps < 0.01$) and R5 was also significantly different from all other repetitions (mean = 0.045, SE = 0.004, all $ps < 0.025$). The main effect of axis-dimension was also significant ($F_{(1,38)} = 55.940$, $p < 0.0001$, $\eta^2 = 0.595$) with the variability on the horizontal dimension being larger (mean = 0.075, SE = 0.005) than the variability on the vertical dimension (mean = 0.054, SE = 0.04). The main effect of group did not reach a significance ($F_{(1,38)} = 1.289$, $p = 0.263$, $\eta^2 = 0.033$) (HV group: mean = 0.060, SE = 0.006; LV Group: mean = 0.069, SE = 0.006). The interaction between repetition and axis-dimension ($F_{(4,152)} = 6.640$, $p < 0.0001$, $\eta^2 = 0.149$) was found to be significant with a Greenhouse-Geisser correction ($\varepsilon = 0.668$). Post-hoc revealed that the variability on the horizontal dimension was significantly higher than the vertical dimension in all 5 repetitions (all *ps < 0.005*). There was also a

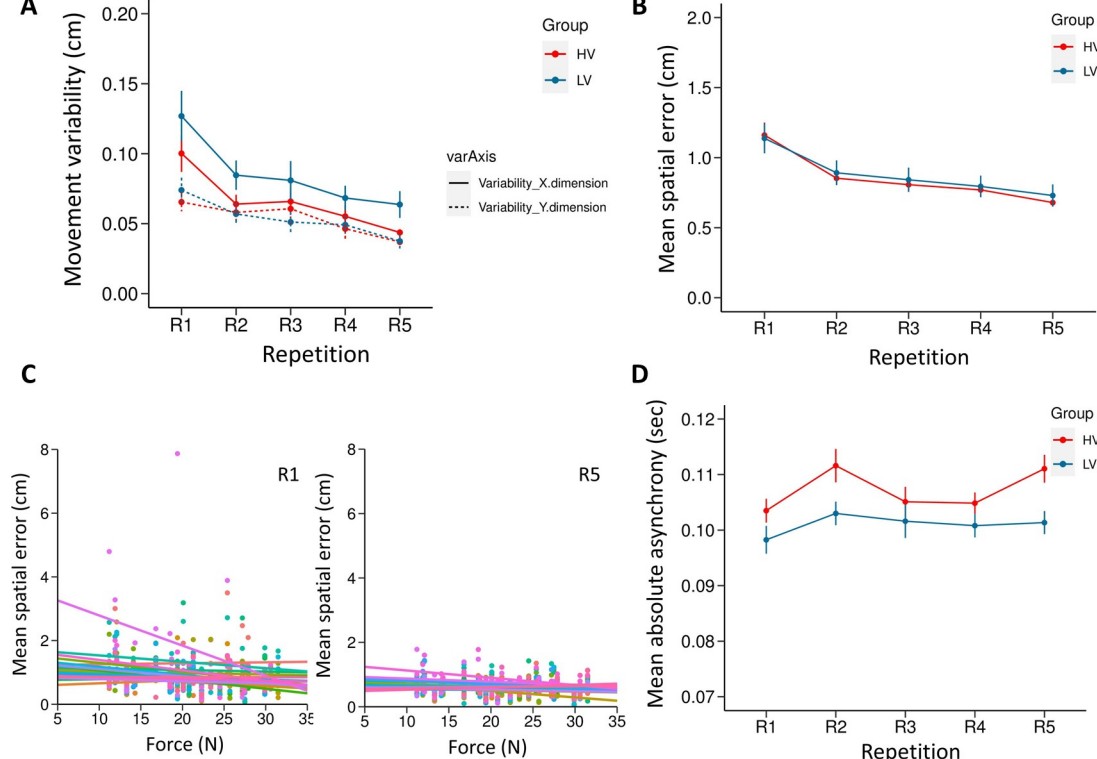

**Fig 7. Results of Experiment 2.** (A) Normalized variability of participant's movement in the horizontal and vertical dimension. (B) Spatial error as a measure of individual performance. The spatial error at each repetition is an average of all 8 configurations from each of the 8 blocks. (C) Repeated measures correlation analysis between the force perturbation experienced and the spatial error of the participants in the HV group, for R1 and R5 separately. Each participant's data and corresponding rmcorr fit lines are shown in different colors. (D) Absolute Asynchronies as a measure of joint performance. All error bars display standard error of mean.

significant interaction between the group and axis-dimension (F(1, 38) = 10.544, p<0.001, $\eta^2$ = 0.595). However, post-hoc analyses did not reveal any significant difference (all *p*s>0.07). All other interactions were non-significant (*p*s > 0.86; see S1 File).

## Spatial error

The ANOVA with a Greenhouse-Geisser correction ($\varepsilon$ = 0.590) revealed a main effect of Repetition, indicating that participants in both groups learned to reduce spatial error over time (F (4, 152) = 46.580, p< 0.0001, $\eta^2$ = 0.551, see Fig 7B). The post-hoc analysis revealed that spatial error at R1 (mean = 1.149, SE = 0.069) was significantly higher compared to R2-R5 (all *p*s<0.0001) and R5 (mean = 0.705, SE = 0.042) was significantly lower compared to R1-R4 (all *p*s<0.001). However, the main effect of group was not significant (F (1, 38) = 0.072, p = 0.789, $\eta^2$ = 0.002; HV group: Mean = 0.854, SE = 0.06; LV group: Mean = 0.879, SE = 0.06). The interaction between the factors was also not significant (F (4,152) = 0.328, p = 0.756, $\eta^2$ = 0.009). The analysis on confederate's spatial accuracy can be found in the (S2 File).

A Bayesian Independent Samples T-Test was performed to assess the likelihood of the spatial accuracy performance between HV and LV groups being similar. The average spatial error across repetitions and learning blocks was computed for both groups and subjected to comparison across the two groups. We tested the null hypothesis, $H_0$: mean spatial error of HV = mean spatial error of LV, and the alternate hypothesis, $H_1$: mean spatial error of HV $\neq$ mean spatial error of LV. The prior is described by a Cauchy distribution centred around zero and a default width parameter of 0.707. The analysis resulted in a Bayes Factor, $BF_{01}$ = 3.146 (see Table 1 for descriptive). This Bayes Factor implies moderate evidence for $H_0$, which means that the data are 3.146 times more likely to occur under $H_0$ than under $H_1$. The Bayes Factor provides moderate evidence in favour of the null hypothesis that the HV and LV groups are not different.

A repeated measures correlation (rmcorr) was computed to determine the relationship between force perturbation and spatial error in the HV group for R1 and R5 separately. The analysis revealed a significant negative correlation between the force perturbation experienced and the spatial error at R1 ($r_{rm}$ (437) = -0.154, p = 0.001; 95% CI = 0.007834378, 0.1892928) and at R5 ($r_{rm}$ (447) = -0.135, DF = 447, p = 0.004; 95% CI = -0.2254212–0.04330164). A scatter plot summarizes the results (see Fig 7C). The results indicate that in the HV group, participants achieved higher accuracy when encountering larger perturbations under conditions of complete predictability, suggesting that they learned to adapt to larger forces by becoming more resilient.

## Absolute Asynchronies

The ANOVA conducted on the absolute asynchronies between the two actors revealed a main effect of Repetition (F (4, 152) = 2.439, p = 0.049, $\eta^2$ = 0.06, see Fig 7D). The post-hoc revealed that R1 (mean = 0.101, SE = 0.002) was significantly lower than R2 (mean = 0.107, SE = 0.002) and R5 (mean = 0.106, SE = 0.002) (all *p*s<0.03), indicating that asynchrony increased over repetitions. The mean asynchrony between the two groups were also found to be significantly

**Table 1. Descriptive statistics.** Descriptives of the comparison of mean spatial error in HV and LV group of Experiment 2, by means of a Bayesian independent t-test.

| Group | N | Mean Spatial Error | SD | SE | 95% Credible Interval | |
|---|---|---|---|---|---|---|
| | | | | | Lower | Upper |
| HV | 20 | 0.854 | 0.192 | 0.043 | 0.764 | 0.944 |
| LV | 20 | 0.879 | 0.370 | 0.083 | 0.706 | 1.053 |

different (F(1,38) = 12.541, p = 0.001, $\eta^2$ = 0.248) with HV group having a higher asynchrony (mean = 0.107, SE = 0.001) compared to the LV group (mean = 0.101, SE = 0.001). The interaction between the factors was not significant (F (4,152) = 0.714, p = 0.583, $\eta^2$ = 0.018).

## Discussion

Analysis of participants movement variability indicate that participants were successfully regulating their own variability by reducing it over repetitions. Participants were generally more variable in their horizontal dimension as in the previous experiment, but differently from Experiment 1, HV group participants were not more variable compared to LV group participants. Also, participants who performed joint actions with a highly variable partner were as accurate as participants who performed joint actions with a less variable partner. Thus, varying force perturbations from a partner was not detrimental to the individual performance. This result indicated that participants in the HV group were able to predict upcoming variable force perturbations and to offset these perturbations. Further support for this interpretation comes from the correlation analysis indicating a negative correlation between the force experienced and the spatial error of participants in the high variability group. The results suggest that participants in the HV group adapted more to larger force perturbations, possibly becoming more resilient under conditions of complete predictability. Taken together, these results suggest that predictability of partner's movement sequence seems to be a pre-condition for potential benefits of individual performance. However, it should be noted that the joint performance in HV group was worse than LV group in the current experiment, as was the case with a non-predictable partner in Experiment 1. Thus, it seems that when the partner is completely predictable, participants can improve the efficiency of their individual performance but not the joint performance.

## Experiment 3

In this experiment we aimed to better understand what needs to be predictable about a partner's movement sequence so that individuals can start to benefit from performing joint actions with a more variable partner. There are two potential properties of force that are relevant in the present task, the magnitude and the direction of the force perturbation exerted by the partner. To tease apart the contributions of these two factors in Experiment 3, we investigated whether partial predictability of a partner's movements is sufficient to achieve high accuracy of performance with a highly variable partner. Here, the partner produced force perturbations that followed a predictable structure in terms of the magnitude of force across the movement sequences, while the direction of force kept changing. Thus, participants could only predict the magnitude of upcoming perturbations but not the direction of the perturbations. If complete predictability of a partner's movements is necessary to offset negative effects of high variability on performance, the pattern of results should be similar to Experiment 1 where neither force nor direction of force perturbations were predictable. If partial predictability is sufficient to reduce the negative impact of highly variable force perturbations, the pattern of results should resemble Experiment 2.

## Materials and methods

### Participants

Forty people participated in the study (26 Females, Mean Age = 26.35 years, SD age = 3.62 years). One participant was later excluded from the analysis due to technical error in the data encoding. All participants were righthanded and reported to have normal or corrected-to-normal vision. Two confederates were hired to participate throughout the study for all 40 participants.

### Ethics statement

The study was approved by the Hungarian ethics committee (United Ethical Review Committee for Research in Psychology- EPKEB). All participants gave informed consent in written form prior to the experiment.

### Procedure

The procedure was the same as in Experiment 2, except for the predictability manipulation in which confederate's movements were made partially predictable. For the participant, magnitude of the force perturbation exerted by the confederate was predictable across repetitions while the direction of the force perturbation changed from repetition to repetition. This was made possible by making the confederate move to a new target location at each repetition of participant's sequence, which changed the direction of the force applied but kept the distance between the target position for the participant and the confederate constant. Hence, in a single block, participants received a single sequence (8 target configurations) that was repeated 5 times, whereas the confederate performed 5 different sequences in which the magnitude of force produced across eight consecutive configurations remained (relatively) constant while the direction of the force changed from repetition to repetition. The standard deviation of the inter-personal distances within every sequence was used to manipulate the variability of confederate's movement across the two groups: it was 11.11 cm for high variability sequences and 1.26 cm for low variability sequences. However, the average inter-personal distances within every sequence in both groups were maintained the same as in the previous experiments (Mean Inter-personal Distance for HV = 52.24 cm and LV = 52.47 cm).

### Data processing and analysis

Participant's and confederates' kinematic data was segmented and analysed following the same protocol as in Experiment 2. A total of 0.02% trials from HV group and 0.14% trials from LV group were removed as outliers. For the repeated measures correlation analysis, a total of 2.247% trials and 2.40% trials were removed as outliers from repetitions R1 and R5 of HV group respectively.

## Results

### Movement variability on horizontal and vertical dimension

The mixed ANOVA revealed a main effect of Repetition, indicating that participants in both groups reduced their variability over time ($F(4,148) = 5.992$, $p<0.0001$, $\eta^2 = 0.139$, see Fig 8A). Post-hoc analysis revealed that the variability at first repetition, R1 (mean = 0.076, SE = 0.005) was significantly higher compared to other repetitions (all $p$s<0.01). The main effect of axis-dimension was also significant ($F(1,37) = 26.637$, $p<0.0001$, $\eta^2 = 0.419$) with the variability on the horizontal dimension being larger (mean = 0.066, SE = 0.004) than the variability on the vertical dimension (mean = 0.055, SE = 0.004). The main effect of group did not reach a significance ($F(1,37) = 1.975$, $p = 0.168$, $\eta^2 = 0.051$; HV group: mean = 0.066, SE = 0.005; LV group: mean = 0.056, SE = 0.005). None of the interactions were significant (all $p$s >0.14; see S1 File).

### Spatial error

The ANOVA with a Greenhouse-Geisser correction ($\varepsilon = 0.667$) revealed a main effect of Repetition, confirming that participants in both groups learned to reduce spatial error over time ($F(4, 148) = 12.164$, $p< 0.0001$, $\eta^2 = 0.247$, see Fig 8B). Post-hoc analysis showed that spatial error at R1 (mean = 1.034, SE = 0.056) was significantly higher compared to R2-R5 (all

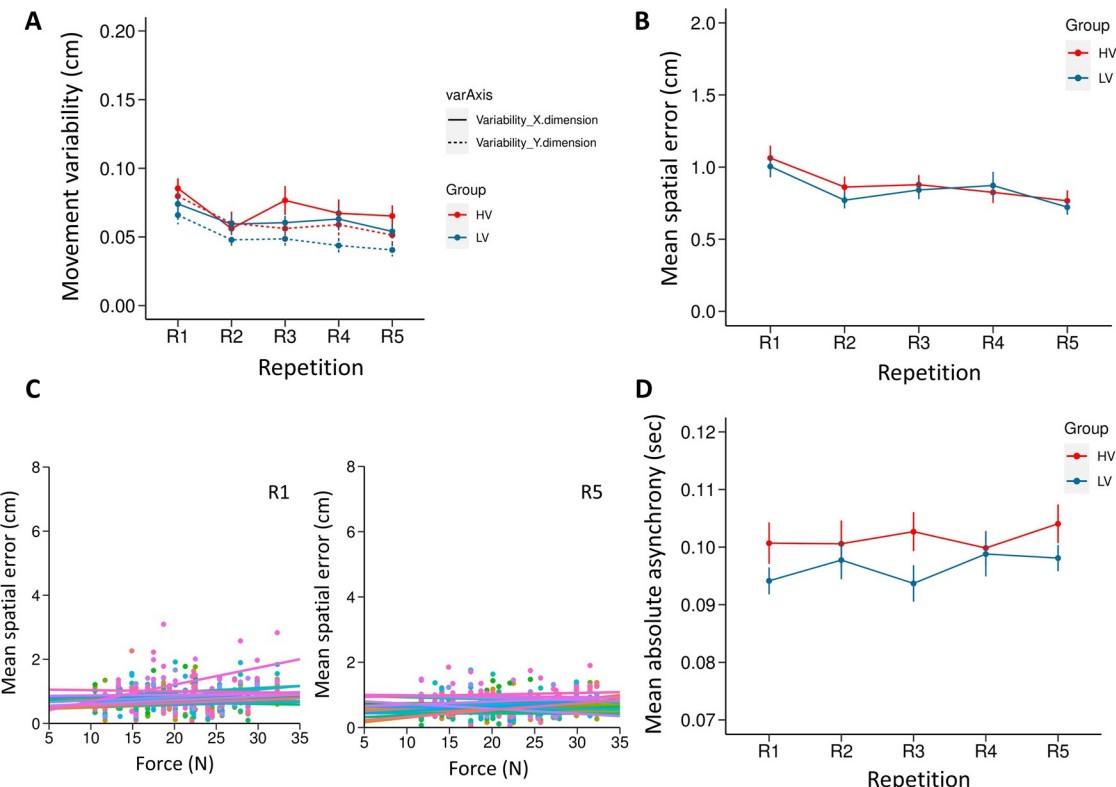

**Fig 8. Results of Experiment 3.** (A) Normalized variability of participant's movement in the horizontal and vertical dimension. (B) Spatial error as a measure of individual performance. The spatial error at each repetition is an average of all 8 configurations from each of the 8 blocks. (C) Repeated measures correlation analysis between the force perturbation experienced and the spatial error of the participants in the HV group, for R1 and R5 separately. Each participant's data and corresponding rmcorr fit lines are shown in different colors. (D) Absolute Asynchronies as a measure of joint performance. All error bars display standard error of mean.

$p$s<0.01). The main effect of Group (HV group: Mean = 0.879, SE = 0.06 and LV group: Mean = 0.842, SE = 0.060) failed to reach a significance (F (1, 37) = 1.179, p = 0.675, $\eta^2$ = 0.005). The interaction between the factors was also not significant (F (4,148) = 0.707, p = 0.534, $\eta^2$ = 0.019). The analysis on confederate's spatial accuracy could be found in the (S2 File).

A Bayesian Independent Samples T-Test was performed to assess the likelihood of performance in the HV and LV group being equal. The average spatial error across each repetition and learning block was computed for both groups and means were subjected to comparison. The null hypothesis, $H_0$: mean spatial error of HV = mean spatial error of LV, was tested against the alternate hypothesis, $H_1$: mean spatial error of HV $\neq$ mean spatial error of LV. The prior is described by a Cauchy distribution centred around zero and a default width parameter of 0.707. The analysis resulted in a Bayes Factor, $BF_{01}$ = 2.986 (see Table 2 for descriptive). This value of the Bayes Factor indicates anecdotal evidence for $H_0$, which means that the data are 2.986 times more likely to occur under $H_0$ compared to $H_1$. The Bayes Factor provides anecdotal evidence in favour of the null hypothesis than the alternative hypothesis.

In addition, a Bayesian Independent Samples T-Test was performed to assess the likelihood of the performance of HV group participants in Experiment 2 and 3 being similar. The average spatial error across each repetition and learning block was computed for HV groups from both experiments and the means were subjected to comparison. The null hypothesis, $H_0$: mean spatial error of HV group in Experiment 2 = mean spatial error of HV group in Experiment 3,

**Table 2. Descriptive statistics.** Descriptives of the comparison of mean spatial error in HV and LV group of Experiment 3, by means of a Bayesian independent t-test.

| Group | N | Mean Spatial Error | SD | SE | 95% Credible Interval | |
|---|---|---|---|---|---|---|
| | | | | | Lower | Upper |
| HV | 19 | 0.879 | 0.278 | 0.064 | 0.745 | 1.013 |
| LV | 20 | 0.842 | 0.260 | 0.058 | 0.720 | 0.964 |

was tested against the alternate hypothesis, $H_1$: mean spatial error of HV group in Experiment 2 $\neq$ mean spatial error of HV group in Experiment 3. The prior is described by a Cauchy distribution centred around zero and a default width parameter of 0.707. The analysis resulted in a Bayes Factor, $BF_{01} = 3.074$ (see Table 3 for descriptive). This value of the Bayes Factor indicates moderate evidence for $H_0$, meaning that the data are 3.074 times more likely to occur under $H_0$ compared to $H_1$. In other words, the Bayes Factor provides more evidence in favour of the null hypothesis that there is no difference in spatial error between the HV groups in Experiment 1 and 2.

A repeated measures correlation (rmcorr) was computed for R1 and R5 separately, to determine the relationship between the force perturbation experienced and the spatial error in the HV group. The analysis revealed a significant correlation between the force perturbation and the spatial error at R1 ($r_{rm}(415) = 0.131$, $p = 0.007$, 95% CI = 0.0355664 0.2247967). However, there was no correlation observed at R5 ($r_{rm}(385) = 0.067$, $p = 0.181$, 95% CI = -0.03217353 0.166803) suggesting that participants learned to adapt to larger forces as in Experiment 2, over the course of training. A scatter plot (see Fig 8C) summarizes the results.

## Absolute asynchronies

The ANOVA conducted on the absolute asynchronies of the two actors did not show a significant main effect of repetition ($F(4,148) = 0.563$, $p = 0.690$, $\eta^2 = 0.015$) indicating that asynchrony between the actors remained the same across repetitions (see Fig 8D). The mean asynchrony between the two groups (HV group: Mean = 0.102, SE = 0.002; LV group: mean = 0.097, SE = 0.002) also did not reach significance ($F(1,37) = 2.406$, $p = 0.129$, $\eta^2 = 0.0061$). The interaction between the factors was also not significant ($F(4,148) = 0.725$, $p = 0.576$, $\eta^2 = 0.019$).

A Bayesian Independent Samples T-Test was performed to assess the likelihood of the absolute asynchrony in the HV and LV group being equal. The average asynchrony across each repetition and learning block was computed for both groups and the means were subjected to comparison. The null hypothesis, $H_0$: mean absolute asynchrony of HV = mean absolute asynchrony of LV, was tested against the alternate hypothesis, $H_1$: mean absolute asynchrony of HV $\neq$ mean absolute asynchrony of LV. The prior is described by a Cauchy distribution centred around zero and a default width parameter of 0.707. The analysis resulted in a Bayes Factor, $BF_{01} = 1.253$ (see Table 4 for descriptive). This value of the Bayes Factor indicates anecdotal evidence for $H_0$, meaning that the data are 1.253 times likely to occur under $H_0$ compared to $H_1$. The Bayes Factor provides anecdotal evidence in favour of the null hypothesis that absolute asynchrony in HV and LV groups are not different.

**Table 3. Descriptive statistics.** Descriptives of the comparison of mean spatial error in HV groups of Experiment 2 and 3, by means of a Bayesian independent t-test.

| Group | N | Mean | SD | SE | 95% Credible Interval | |
|---|---|---|---|---|---|---|
| | | | | | Lower | Upper |
| Experiment 2 | 20 | 0.854 | 0.192 | 0.043 | 0.764 | 0.944 |
| Experiment 3 | 19 | 0.879 | 0.278 | 0.064 | 0.745 | 1.013 |

**Table 4. Descriptive statistics.** Descriptives of the comparison of mean absolute asynchrony in HV and LV groups by means of a Bayesian independent t-test.

| | Group | N | Mean | SD | SE | 95% Credible Interval | |
|---|---|---|---|---|---|---|---|
| | | | | | | Lower | Upper |
| Absolute Asynchrony | HV | 19 | 0.102 | 0.011 | 0.002 | 0.096 | 0.107 |
| | LV | 20 | 0.097 | 0.009 | 0.002 | 0.092 | 0.101 |

## Discussion

As in Experiment 2, participants successfully regulated the variability of their own movements over time It was also observed that participants were more variable on the horizontal axis, which was less constrained by haptic coupling, owing to wider action exploration on this dimension. There was also no indication of a difference in the participant's individual performance or learning between the high and the low variability group. The correlation analysis suggests that while the force perturbations predicted the spatial error in the first repetition, participants became equally accurate for large force perturbations as for smaller force perturbations in the last repetition, despite the direction of the force perturbations being unpredictable. Thus, partial predictability of the force perturbations of the partner's movement sequence was sufficient to offset the negative effects of high magnitude of partner's variability. Also, contrary to Experiment 2, there was no main effect of group for the joint performance, measured by the temporal asynchronies, when partners movements were only partially predictable. Our results indicate that, when interacting with a partner who is only partially predictable but highly variable, participants seems to improve both their individual and joint performances over time.

## General discussion

During joint action coordination that involves physical coupling, motor variability produced by an actor can have a direct impact on the partner's movements. In the current study, we investigated whether individuals could regulate their own and their partner's variability when learning a motor task together. Specifically, we looked at how variability at different degrees of predictability of force perturbations coming from a partner, during a joint action, foster or hamper individual and joint performance. Furthermore, we aimed at identifying what strategies individuals can adopt to benefit both individual and joint performance. Motor learning generally involves reduction of variability or error in one's movement over practice. Action exploration and subsequent exploitation of the best explored actions are found to be aiding this learning process in specific types of learning. In our study, participants could either utilize their own variability, their partner's variability, or both, to explore possible motor refinements that allow successful learning.

The finding on participants' movement variability showed that in all three experiments, participants were generally more variable on the dimension that was less constrained by haptic coupling, that is the horizontal dimension. Also, participants learned to reduce their movement variability and consequentially, their spatial error over time in all three experiments. We suggest that regardless of the experimental condition, individuals utilised their internal variability to explore their motor parameters relevant for the task which benefitted their individual performance. However, individuals might adopt different strategies when it comes to utilizing external variability coming from their partner during a joint action.

In Experiment 1, where partner's variability was unpredictable, individuals learned to reduce their spatial error over time, but HV group participants were worse than LV group in their performance. Also, the correlation analysis showed that when the partner's movements were not predictable, individuals were unable to compensate for large perturbations. These results

suggest that the high variability of the partner was not providing any task-relevant information as participants' movements were perturbed in random directions. In other words, the partner's variable and unpredictable movements had a detrimental effect on individual performance. However, the fact that participants learned to reduce their spatial error over time points to the possibility that they could resort to another strategy, i.e. the modulation of their own variability to ensure exploration and improvement in performance. In Experiment 2 and 3, where the partner's movements were completely or at least partially predictable, spatial accuracy was not impacted. Importantly, it was observed that individuals learned to compensate for a wide range of force perturbations when the partner's movements were partially predictable and even more when they were completely predictable, suggesting that they developed resilience to larger perturbations. These results indicate that predictability becomes a necessary pre-condition for partner's variability to positively influence individual's performance. In Experiment 2 and 3, the improvements in individual performance was possibly aided by both regulation of their own movement variability as shown by the variability analysis as well as their partner's variability. Taken together the results on individual performances of the three experiments, it seems that it is not the magnitude of variability, rather the structure of task-relevant variability coming from the partner that supports individual learning and action exploration.

We propose that under conditions of predictability, a varied range of force perturbations experienced from a partner allows individuals to generate a flexible internal model accommodating various movement solutions for a wider range of task parameters (in our case, various force perturbations) [21]. Thus, in the high-variability condition, instead of forming a single motor plan required to perform the task under a constant force, individuals generated multiple motor plans for performing the task under a varying task environment. Depending on the perturbation experienced, the most optimal plan could be exploited for improving the efficiency of outcome achievement. This allowed them to exhibit similar performance as in the low-variability condition (Experiment 2 and 3). Had a wider action exploration not been fostered by partner's variability in these cases, participants in the high-variability group should have performed worse than the low variability group due to the more varied force perturbations.

It is important to note that the individual performance does not necessarily reflect the success of joint performance. The results of the analyses on joint performance show that participants in the HV group in Experiment 1 and 2 were worse than participants in the LV group at synchronising with their partners. As the asynchrony increased while the individual spatial accuracy decreased over repetitions, our results indicate that individuals traded off the spatial accuracy and the temporal coordination in Experiment 1 and 2. This suggests that individuals failed to accommodate high variability coming from a partner to benefit the joint performance. On the other hand, the results of individual movement performance show a different pattern, as mentioned before. While the partner's high variability seemed to hamper performance in Experiment 1, the partners high variability produced in a predictable manner in Experiment 2 allowed individuals to explore their action space and gather motor information relevant for the task. This was reflected in a benefit of individual performance. The differential effects seen in the individual and joint performances in Experiment 2 is rather interesting. If anything, one would have expected the joint performances of both groups to benefit from the complete predictability of the partner's movements as conditions for an optimal joint performances are hard-wired in this experiment (due to *complete predictability* of partner's movements). This should have allowed individuals to adopt predictability as a coordination strategy for improving their joint performance. However, it seems that under such optimal conditions for joint action, a higher variability of the partner's movements (as in the HV group) only interferes in successful reliance on predictability as a coordination strategy. In other words, it is reasonable to assume that individuals will benefit from such predictability within the joint action context only when partner's kinematics are the least variable.

In Experiment 3, even though results of the Bayesian analysis only provides anecdotal evidence for no difference in the temporal asynchronies between the two groups, one could speculate that when the partner's variability was high and the perturbations were *partially* predictable, participants may have exploited action exploration strategies to improve their joint performance. It is important to note that, unlike in Experiment 2, optimal conditions for joint action were not hard-wired in Experiment 3, i.e., the partner's movements are only *partially predictable*. It is possible that the variability coming from the partner's movement provided some room for parameters exploration that could be exploited by participants to support the performance of not only their individual action, but also the joint action.

Our findings indicate that individuals might modulate their action exploration strategy by selectively relying on either their own or on the partner's variability to improve their individual performance, joint performance or both- depending upon the structure of variability provided to them by their partner.

One potential limitation of the current paradigm is the lack of a test phase following the training phase which would have allowed us to measure how the training contributes to long term retention of the learned skill or the transfer of the learned skill to a different movement context. However, the extensive training sessions over multiple movement sequences, in all three experiments, provides actors with sufficient time to reach an asymptote or saturation of individual performance, as can be observed from all the spatial error analyses. Additionally, it was observed that participants offset potential speed-accuracy trade off in performance (see S4 File, for movement time analysis), which is a characteristic of skill learning. Thus, even though we do not capture long-term retention with the current experimental design, our data indicate that individual and joint action learning took place within the experimental sessions [22, 23]. A potential future research direction could be to investigate how the individual and joint performances will differ if both actors hold symmetrical knowledge about the task and test the contribution of high variability training on long term retention of the learned skill.

## Supporting information

**S1 File. Interaction effects from variability analysis.**
(PDF)

**S2 File. Analysis on confederate's data.**
(PDF)

**S3 File. Repeated measures correlations (rmcorr) on LV group data.**
(PDF)

**S4 File. Movement time analysis.**
(PDF)

**S5 File. Miscellaneous analysis.**
(PDF)

**S6 File. Data.**
(XLSX)

## Acknowledgments

The authors would like to thank Neeraj Kumar for his input in programming of the experiments.

## Author Contributions

**Conceptualization:** Simily Sabu, Arianna Curioni, Cordula Vesper, Natalie Sebanz, Günther Knoblich.

**Data curation:** Simily Sabu.

**Formal analysis:** Simily Sabu, Arianna Curioni, Günther Knoblich.

**Funding acquisition:** Natalie Sebanz, Günther Knoblich.

**Methodology:** Simily Sabu, Arianna Curioni, Cordula Vesper, Natalie Sebanz, Günther Knoblich.

**Project administration:** Günther Knoblich.

**Supervision:** Arianna Curioni, Günther Knoblich.

**Validation:** Günther Knoblich.

**Visualization:** Simily Sabu.

**Writing – original draft:** Simily Sabu.

**Writing – review & editing:** Simily Sabu, Arianna Curioni, Cordula Vesper, Natalie Sebanz, Günther Knoblich.

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
