## [Decision Letter · Decision Letter 0]

21 Apr 2020

PONE-D-20-03780

How does a partner’s motor variability affect joint action?

PLOS ONE

Dear Ms Sabu,

Thank you for submitting your manuscript to PLOS ONE. After careful consideration, we feel that it has merit but does not fully meet PLOS ONE’s publication criteria as it currently stands. Therefore, we invite you to submit a revised version of the manuscript that addresses the points raised during the review process.

We would appreciate receiving your revised manuscript by Jun 05 2020 11:59PM. To enhance the reproducibility of your results, we recommend that if applicable you deposit your laboratory protocols in protocols.io, where a protocol can be assigned its own identifier (DOI) such that it can be cited independently in the future. For instructions see: http://journals.plos.org/plosone/s/submission-guidelines#loc-laboratory-protocols

We look forward to receiving your revised manuscript.

Kind regards,

Gavin Buckingham

Academic Editor

PLOS ONE

Journal Requirements:

2. Please change "female” or "male" to "woman” or "man" as appropriate, when used as a noun.

3. We note that Figure 1 includes an image of a participant. 

Reviewers' comments:

Reviewer's Responses to Questions

**Comments to the Author**

1. Is the manuscript technically sound, and do the data support the conclusions?

Reviewer #1: Yes

Reviewer #2: Partly

2. Has the statistical analysis been performed appropriately and rigorously? 

Reviewer #1: Yes

Reviewer #2: Yes

3. Have the authors made all data underlying the findings in their manuscript fully available?

Reviewer #1: Yes

Reviewer #2: Yes

4. Is the manuscript presented in an intelligible fashion and written in standard English?

Reviewer #1: Yes

Reviewer #2: Yes

5. Review Comments to the Author

Reviewer #1: The paper addresses an interesting issue regarding which factors modulate the efficiency of learning to perform a joint-task. Specifically, in three experiments the authors tested whether the partner’s motor variability and the predictability of her movement perturbations influence the participants’ performance.

The paper is interesting, technically sound and well written, although some methodological improvements could be implemented and some clarifications are needed. Please see my comments below.

EXPERIMENT 1.

1)Figure 4. Does it show the performance of a participant during the individual or the joint phase?

2)Why the participants’ performance in the individual phase was not used to normalize the dependent variables in the joint phase? This would rule out the possibility that the main effect of group was due to a random group-difference in the participant’s ability to perform the task, which is still a possible alternative explanation, given the between-group nature of the design.

3)Line 316: “trials in which participants moved to positions 5 cm or more away outside the target grid were considered as outliers and removed from the analysis”. How many trials were removed? Were they equally distributed between the groups?

4)Line 319-320: “The spatial error for all 10 repetitions of the sequences was calculated and averaged across all 8 training blocks”. First, I would remove the term “training” here as it is confusing. Second, do the authors mean that they averaged the dependent variables between the 8 configurations of each repetition of each sequence within each block, and then averaged the values between the blocks?

5)With regard to the correlation analysis (correlation between force perturbation and spatial error), I do not understand how it was performed. I would have expected the authors to use the information of each single trial, and to examine whether within each participant the trials in which there is higher perturbation they also show higher errors. This should be done by transforming (Z-transformation) the dependent variable within each repetition (to control for the main effect of repetition) and running a linear mixed model to take into account inter-individual variability.

Please note that these five comments stand for Experiment 2 and 3 as well.

6)Line 320-321: “Mean Spatial Error of Groups were also subjected to a Bayesian Independent Samples T-Test for comparisons.” I do not see the results of the Bayesian test in Experiment1, this might be a typo.

7)With regard to the results on Spatial error and Individual Variability on horizontal dimension, the ANOVA showed an interaction effect: the authors should then report the post-hoc tests and specify which repetitions showed a significant between-group difference. For instance, it seems that R2 showed no group difference, how do the authors interpret this result? Please comment.

8)Minor: line 272, delete the “s”.

EXPERIMENT 2.

1)I do not understand why the participants performed 10 repetitions in Experiment 1 and only 5 repetitions in Experiment 2. Could you please clarify this point?

2)The authors may want to compare the three dependent variables between the HV groups in Experiment1 and Experiment2, to show that indeed the predictability of the partner’s movements allows a faster learning in the HV group in Experiment 2. Please note that the expected result here is that the two groups do not differ in the first repetition, and then improve faster.

EXPERIMENT 3.

1)As for Experiment 2, the authors may want to compare the three dependent variables between the HV groups in Experiment 1, 2 and 3, to show that indeed the predictability of the partner’s movements allows a faster learning in the HV group in Experiment 2 and 3. Please note that the expected result here is that the three groups do not differ in the first repetition, and then improve faster (and equally) in Experiment 2 and 3.

2)Similarly, the authors may want to run the same Bayesian test that they did in Experiment 2 and 3 to compare the average spatial error between HV groups in Experiment 2 and 3.

Reviewer #2: I have reviewed the paper “How does a partner’s motor variability affect joint action” by Sabu et al. The paper examines the issue of “joint learning” between two individuals, looking at the role of variability and predictability. Using a sequence learning paradigm in 3 experiments, the authors report that higher levels of unpredictable forces result in decreased performance (Exp 1), the same levels of forces when forces are predictable eliminate this decrement (Exp2), and that making the forces ‘partially predictable’ (magnitude but not direction) also eliminates the decrement found in Exp 1.

I think the strengths of the paper are its novel experimental setup and relatedly elegant experimental manipulations. The methods were also well done for the most part. However, I do have some major concerns with the analyses and the underlying theoretical motivation

Major concerns:

1.The argument about ‘predictability’ in Exp 2 and Exp 3 requires some work in my opinion. Since the participant did not have “apriori” knowledge of the confederate’s sequence and could only discover the fact that it was predictable over practice, it seems surprising that the differences found in Experiment 1 disappear in even in the very first block (R1). How could participants predict the confederate’s sequence so quickly? In my view it is essential to show this first block in more ‘fine-grained’ resolution. So I would suggest not averaging all 10 sequences within the first block and instead show this individually (presumably for the first 1-3 sequences, the results should look exactly the same as Exp 1).

2.The analysis of ‘horizontal’ and ‘vertical’ variability separately does not make sense in the task given that there was no primary direction of motion. These could be lumped in as a bivariate variable error computed along both dimensions simultaneously.

3.Ln 248 says that the task required “synchronous” matching with the partner– yet no information is provided about the temporal accuracy with learning. Relatedly the movement times also need to be provided to make sure that interpretation of changes in accuracy are not confounded by speed-accuracy tradeoffs.

4.Some measure of performance on the ‘confederate’s side would also be critical to understand the performance reported here – since this is a person and not a robot, were there adjustments that the confederate made over time, or in different conditions that potentially affected the participant’s learning?

5.Finally, I felt that the authors’ underlying theoretical motivation needs to be explained better in terms of the task used here. The authors refer to Wu et al and exploration, but the tasks in Wu et al (shape matching and force field adaptation) *require* exploration to move from one movement pattern to another. Here in the sequence learning paradigm, it is not clear *why* exploration is required to perform this task (and therefore why variability should be useful). Instead, the changes reported in accuracy/precision are simply refinements of an existing movement pattern. The authors need to address this more carefully both in the Intro and the Discussion and mention why exploration is required for learning this sequence task.

Minor:

1.Ln 28 – “individual motor learning studies demonstrate individual’s ability//”. The term “individual studies” is confusing

2.Ln 58 “in such joint actions and in joint actions” – rephrase

3.Ln 65 – “making one’s own movement less variable and more consistent”. Is less variable the same as more consistent or does it refer to some other feature here?

4.Ln 76 – the paragraph on interpersonal coordination seems a little out of place here. Either develop this more fully or move it to the Discussion if it is not central to the introduction

5.Ln 95-98: The study referred here actually showed that the magnitude of variability was not important – rather it was the structure of the variability (as measured by the autocorrelation) that determined learning rate.

6.Ln 103-104: The two statements here referring to [19,20] need to be explained in a bit more detail to help understand why variability may play multiple roles.

7.Ln 118-130 seems to be talking about generalization – yet this is not a major focus of the paper.

8.Ln 139-142. The two options spelled out here are interesting, but do not seem to be tested in the paper at all

9.Ln 230 – remove the “/sec” - just 120 Hz

10.In all Experiments, was any feedback provided to the participants (in terms of errors or times etc.)?

11.Were data from the “individual action phase” analyzed at all – it seems like this would be important to understand how the learner changed strategies in the joint learning task

12.Ln 345 – please report exact p-values unless they are very small (Say <.001). Same comment holds for other part of the manuscript as well.

13.Ln 358-365: The number of data points on Fig 5B seem larger than the total number of participants in the HV group.

14.Ln 365 – Not sure what this means. The author say that the LV group experienced variation (just lower in magnitude) in Ln 257– so why is the LV not plotted here?

15.Ln 363 – not clear what the authors mean by “correlation is significant even without removing the outliers”. If they are truly outliers, they need to be removed because they will affect the magnitude of the correlation (even if the significance is not affected)

16.Ln 455 – please use full sentences to describe this.

17.At several places, the authors seem to go back and forth between the terms ‘Variability’ and “Group’ to describe the between subject factor – in my view ‘Group’ is much clearer since there are only two groups.

6. PLOS authors have the option to publish the peer review history of their article (what does this mean?). If published, this will include your full peer review and any attached files.

Reviewer #1: Yes: Lucia Maria Sacheli

Reviewer #2: No

---

## [Author Response · Author response to Decision Letter 0]

5 Jul 2020

Journal Requirements

Reply: Done.

2. Please change "female” or "male" to "woman” or "man" as appropriate, when used as a noun.

Reply: Necessary changes have been made. 

3. We note that Figure 1 includes an image of a participant. As per the PLOS ONE policy (http://journals.plos.org/plosone/s/submission-guidelines#loc-human-subjects-research) on papers that include identifying, or potentially identifying, information, the individual(s) or parent(s)/guardian(s) must be informed of the terms of the PLOS open-access (CC-BY) license and provide specific permission for publication of these details under the terms of this license. Please download the Consent Form for Publication in a PLOS Journal (http://journals.plos.org/plosone/s/file?id=8ce6/plos-consent-form-english.pdf). The signed consent form should not be submitted with the manuscript but should be securely filed in the individual's case notes. Please amend the methods section and ethics statement of the manuscript to explicitly state that the patient/participant has provided consent for publication: “The individual in this manuscript has given written informed consent (as outlined in PLOS consent form) to publish these case details”. If you are unable to obtain consent from the subject of the photograph, you will need to remove the figure and any other textual identifying information or case descriptions for this individual.

Reply: We have obtained and stored relevant confirmations from all depicted individuals.

Reply: We are willing to provide repository information for our data. 

Reviewers' comments:

Reviewer's Responses to Questions

Comments to the Author

1. Is the manuscript technically sound, and do the data support the conclusions?

Reviewer #1: Yes

Reviewer #2: Partly

2. Has the statistical analysis been performed appropriately and rigorously?

Reviewer #1: Yes

Reviewer #2: Yes

 3. Have the authors made all data underlying the findings in their manuscript fully available?

The PLOS Data policy requires authors to make all data underlying the findings described in their manuscript fully available without restriction, with rare exception (please refer to the Data Availability Statement in the manuscript PDF file). The data should be provided as part of the manuscript or its supporting information or deposited to a public repository. For example, in addition to summary statistics, the data points behind means, medians and variance measures should be available. If there are restrictions on publicly sharing data—e.g. participant privacy or use of data from a third party—those must be specified.

Reviewer #1: Yes

Reviewer #2: Yes

4. Is the manuscript presented in an intelligible fashion and written in standard English?

Reviewer #1: Yes

Reviewer #2: Yes

5. Review Comments to the Author

Reviewer #1

The paper addresses an interesting issue regarding which factors modulate the efficiency of learning to perform a joint-task. Specifically, in three experiments the authors tested whether the partner’s motor variability and the predictability of her movement perturbations influence the participants’ performance. The paper is interesting, technically sound and well written, although some methodological improvements could be implemented, and some clarifications are needed. Please see my comments below.

EXPERIMENT 1.

1) Figure 4. Does it show the performance of a participant during the individual or the joint phase?

Reply: Figure 4 shows the performance of a participant during the joint phase. Individual phase data is not shown in the manuscript as this data does not capture a baseline performance for the experiment. This point has been detailed in our reply to Reviewer 1 Pt.2.

2) Why the participants’ performance in the individual phase was not used to normalize the dependent variables in the joint phase? This would rule out the possibility that the main effect of group was due to a random group-difference in the participant’s ability to perform the task, which is still a possible alternative explanation, given the between-group nature of the design.

Reply: We chose not to use the individual-phase data to normalize the dependent variables in the joint-phase because the former data does not reflect a baseline performance for the latter. Individual-phase data does not include a non-variable force component that participants had to adapt to, only in which case, it could be considered as a baseline performance to the joint-phase. In our experiment, the individual-phase only exists for practice purpose, so that the participants are exposed to the target locations at each Block, without experiencing force perturbations. 

3) Line 316: “trials in which participants moved to positions 5 cm or more away outside the target grid were considered as outliers and removed from the analysis”. How many trials were removed? Were they equally distributed between the groups?

Reply: Outlier information has been added to the revised manuscript.

4) Line 319-320: “The spatial error for all 10 repetitions of the sequences was calculated and averaged across all 8 training blocks”. First, I would remove the term “training” here as it is confusing. Second, do the authors mean that they averaged the dependent variables between the 8 configurations of each repetition of each sequence within each block, and then averaged the values between the blocks?

Reply: Thank you for pointing out that our use of terminology is confusing. We revised the manuscript and replaced all usage of the term ‘training block’ to ‘block’. 

 Yes, the spatial error analysis is done by averaging all the 8 configurations of each sequence taken from the 8 blocks calculated for each of the 10 repetitions. This would mean that in general, the value at every repetition will be an average of 64 data points (8 configurations x 8 blocks). This line has been revised for better readability and it reads as follows (pg. 18, line 400): “The mean spatial error across sequences in all 8 blocks was calculated for each of the 10 repetitions of the sequences and was subjected to a mixed 2x10 ANOVA with Group (HV and LV) as the between-subject factor and Repetitions (R1-R10) as the within-subject factor.”

5) With regard to the correlation analysis (correlation between force perturbation and spatial error), I do not understand how it was performed. I would have expected the authors to use the information of each single trial, and to examine whether within each participant the trials in which there is higher perturbation they also show higher errors. This should be done by transforming (Z-transformation) the dependent variable within each repetition (to control for the main effect of repetition) and running a linear mixed model to take into account inter-individual variability. Please note that these five comments stand for Experiment 2 and 3 as well. 

Reply: We attempted to perform the linear mixed model; however, the analysis could not be performed due to insufficient data. To see if the correlations varied from repetition to repetition, we performed the correlation analysis on each of the repetitions separately. There was no indication that the correlations systematically differed across repetitions, or from the correlations based on the grand average values of spatial error. 

 Also, to address the concern about inter-individual variability, we performed similar analysis across different repetitions on the standardized spatial error values (z-scores calculated using each participant’s mean and SD). There was no difference observed in this case as well. Along with this, correlations were performed on the mean z-scores, like in the original analysis, and there was no indication of a different result. If inter-individual variability was influencing the results, we should have observed a different trend between the correlations on the grand averaged spatial error and the grand-averaged z-scores. We have included the r values for each of these analyses mentioned, in the supporting information (S3 File).

6)Line 320-321: “Mean Spatial Error of Groups were also subjected to a Bayesian Independent Samples T-Test for comparisons.” I do not see the results of the Bayesian test in Experiment 1, this might be a typo.

Reply: This was a typo and we have deleted it in the revised version.

7) With regard to the results on Spatial error and Individual Variability on horizontal dimension, the ANOVA showed an interaction effect: the authors should then report the post-hoc tests and specify which repetitions showed a significant between-group difference. For instance, it seems that R2 showed no group difference, how do the authors interpret this result? Please comment.

Reply: The significant interaction between the group and repetition factor, indeed comes from the difference in performance at R1 (mean (HV)= 1.1706, SE= 0.152; mean (LV)= 1.229, SD= 0.152). It should be noted that interaction was found significant only for the horizontal dimension and not for the vertical dimension. This points to the fact that participants have more degrees of freedom to explore their motor parameters on the horizontal dimension as they have been physically coupled and hence perturbed by the partner, on the vertical dimension. The higher variability seen for HV participants in the initial phase of learning can thus be understood as participants exploring their action space in search of better movement solutions. However, an absence of difference in performance already at R2 shows that participants in the HV group learned to adapt faster, thus learning to be more consistent in their movements for the benefit of their expected outcome. 

 This said, the analysis under question has been removed from the manuscript as we agree with Reviewer 2’s suggestion (see Pt.2) about treating the horizontal and vertical dimensions as a bivariate in the analysis of movement variability. We have instead reported a bivariate analysis for the same. Please see our reply to Reviewer 2 Pt.2 for the analysis. 

8) Minor: line 272, delete the “s”.

Reply: Done. 

EXPERIMENT 2.

1) I do not understand why the participants performed 10 repetitions in Experiment 1 and only 5 repetitions in Experiment 2. Could you please clarify this point?

Reply: Experiment 1 results showed that individuals do not need 10 repetitions to learn the task as they did not improve further after the 5th repetition. We understand that this limits our possibility of comparing results across experiments (see Pt.2) but reducing the number of repetitions also served to react to complaints from our participants. Some of the complained about too much physical exertion in Experiment 1 where the experimental sessions lasted 35 minutes, in which actors had to act against strong force perturbations coming from their partner. Furthermore, we wanted to make sure that there were no performance decrements across repetitions due to fatigue.

2) The authors may want to compare the three dependent variables between the HV groups in Experiment1 and Experiment2, to show that indeed the predictability of the partner’s movements allows a faster learning in the HV group in Experiment 2. Please note that the expected result here is that the two groups do not differ in the first repetition, and then improve faster.

Reply: We agree that this would be an interesting comparison. However, Experiment 1 and 2 cannot be directly compared as the two experiments have different numbers of repetitions. Experiment 1 will have a general performance benefit due to more practice and this would bias the analysis. The possibility of comparing only the first 5 repetitions for Experiment 1 also will not hold, as the joint phase follows a blocked design, which means that every block is preceded by the 10 repetitions from the previous block. 

EXPERIMENT 3.

1) As for Experiment 2, the authors may want to compare the three dependent variables between the HV groups in Experiment 1, 2 and 3, to show that indeed the predictability of the partner’s movements allows a faster learning in the HV group in Experiment 2 and 3. Please note that the expected result here is that the three groups do not differ in the first repetition, and then improve faster (and equally) in Experiment 2 and 3.

Reply: Please see our reply to the previous two points with respect to the comparison between Experiment 1 and the other experiments. 

2) Similarly, the authors may want to run the same Bayesian test that they did in Experiment 2 and 3 to compare the average spatial error between HV groups in Experiment 2 and 3.

Reply: We have now added the Bayesian Independent Samples T-test comparing the HV group in Experiment 2 and 3. The analysis report is as follows (pg. 39): 

“In addition, a Bayesian Independent Samples T-Test was performed to assess the likelihood of the performance of HV group participants in Experiment 2 and 3 being similar. The average spatial error across each repetition and learning block was computed for HV groups from both experiments and means were subjected to comparison. The null hypothesis, H0: mean spatial error of HV group in Experiment 2= mean spatial error of HV group in Experiment, was tested against the alternate hypothesis, H1: mean spatial error of HV group in Experiment 2 ≠ mean spatial error of HV group in Experiment The analysis resulted in a Bayes Factor, BF10 = 0.325 (see Table 3 for descriptive).This value of the Bayes Factor indicates moderate evidence for H0, which means that the data are only 0.325 times more likely to occur under H1 compared to H0. In other words, the Bayes Factor provides more evidence to accept the null hypothesis that there is no difference in spatial error between the HV groups in Experiment 1 and 2.”

 95% Credible Interval

 Group N Mean SD SE Lower Upper

 Experiment 2 20 0.854 0.192 0.043 0.764 0.944 

 Experiment 3 19 0.879 0.278 0.064 0.745 1.013 

Table 3. Descriptive statistics: Descriptives of the comparison of mean spatial error in HV groups of Experiment 2 and 3, by means of a Bayesian independent t-test.

Reviewer #2

I have reviewed the paper “How does a partner’s motor variability affect joint action” by Sabu et al. The paper examines the issue of “joint learning” between two individuals, looking at the role of variability and predictability. Using a sequence learning paradigm in 3 experiments, the authors report that higher levels of unpredictable forces result in decreased performance (Exp 1), the same levels of forces when forces are predictable eliminate this decrement (Exp2), and that making the forces ‘partially predictable’ (magnitude but not direction) also eliminates the decrement found in Exp 1.

I think the strengths of the paper are its novel experimental setup and relatedly elegant experimental manipulations. The methods were also well done for the most part. However, I do have some major concerns with the analyses and the underlying theoretical motivation

Major concerns:

1. The argument about ‘predictability’ in Exp 2 and Exp 3 requires some work in my opinion. Since the participant did not have “apriori” knowledge of the confederate’s sequence and could only discover the fact that it was predictable over practice, it seems surprising that the differences found in Experiment 1 disappear in even in the very first block (R1). How could participants predict the confederate’s sequence so quickly? In my view it is essential to show this first block in more ‘fine-grained’ resolution. So, I would suggest not averaging all 10 sequences within the first block and instead show this individually (presumably for the first 1-3 sequences, the results should look exactly the same as Exp 1).

Reply: We agree that looking at the blocks separately is important. Before we address the suggestion, we would like to clarify our terminology. Please note that the ‘repetitions’ (R1, R2, R3 and so on) are not the same as the ‘block’. In our study, we have 8 blocks in total- each corresponding to a sequence (combination of 8 targets) that the participant needs to repeat (10 times in Experiment 1: R1-R10 and 5 times in Experiment 2 & 3: R1-R5). While the participant receives only one sequence per block, the confederate may receive one or different sequences at each of the repetition, depending on the predictability manipulation (Experiment 1, 2 and 3). The averaging at each of the repetition, for example at R1, is across all 8 target locations of each of the sequences across 8 Blocks, i.e., 64 data points per repetition. 

 This said, we agree that the group difference in the spatial error at R1 for Experiment 1 vs. Experiment 2 and 3 indeed could be due to the block effect. Looking at the fine-grained resolution at Block 1 is relevant for this reason. Please find the results below: 

Experiment 1 : “The ANOVA on the spatial error at only Block 1, with a Greenhouse-Geisser correction (ε =0.578), revealed a main effect of the repetitions, indicating that participants in both groups learned to reduce spatial error over repetitions even within Block 1 (F (9,306) = 26.852, p<0.0001, η2 = 0.441). The post-hoc tests revealed that R1 (mean = 2.335, SE= 0.169) was significantly different from R2-R10 (all ps<0.001). The analysis also showed a significant main effect of group, with the HV group (Mean=1.406, SE= 0.101) having significantly larger errors than the LV group (Mean =1.080, SE =0.101), (F(1, 34) =5.236, p=0.028, η2 = 0.133). The interaction between the two factors was not significant (F (9, 306) = 0.986, p= 0.451 η2 = 0.028).”

Experiment 2: “The ANOVA on the spatial error at Block 1, with a Greenhouse-Geisser correction (ε =0.526) revealed a main effect of Repetition, indicating that participants in both groups learned to reduce spatial error over time (F (4, 152) = 24.316, p< 0.0001, η2 = 0.390).The post-hoc analysis revealed that effect was driven by R1 (mean= 2.215, SE= 0.237) whose spatial error was significantly larger than R2-R5 (all ps<0.001). There was no significant difference between the mean spatial error of HV group (Mean= 1.197, SE= 0.140) and LV group (Mean= 1.234, SE= 0.136) (F (1, 38) = 0.033, p = 0.856, η2 = 0.001). The interaction between the factors was also not significant (F (4,152) = 1.263, p= 0.287, η2 = 0.032).”

Experiment 3: “The ANOVA on the spatial error at Block 1, with a Greenhouse-Geisser correction (ε =0.684) revealed a main effect of Repetition, confirming that participants in both groups learned to reduce spatial error over time (F (4, 148) = 23.196, p< 0.0001, η2= 0.385). The post-hoc revealed that spatial error at R1 (mean = 1,614, SE= 0.129) was significantly higher compared to R2-R5 (all ps<0.001). There was no main effect of Group (F (1, 37) = 0.194, p = 0.662, η2 = 0.005), indicating that spatial error did not differ significantly between HV group (Mean= 1.106, SE= 0.091) and LV group (Mean= 1.072, SE= 0.088). The interaction between the factors was also not significant (F (4,148) = 0.662, p = 0.564, η2 = 0.018).”

The results show that in Experiment 2 and 3, the main effect is absent already at Block 1. We would like to argue that this could be because the pre-exposure to the target locations in the individual phase was enough for the participants to offset the group effect while learning the task together with a partner producing atleast partially predictable movements. Whereas, a highly variable and unpredictable partner only hinders the participants to utilize the task knowledge they gained during individual phase, in Experiment1. 

2.The analysis of ‘horizontal’ and ‘vertical’ variability separately does not make sense in the task given that there was no primary direction of motion. These could be lumped in as a bivariate variable error computed along both dimensions simultaneously.

Reply: Participant’s movement variability was analysed separately on the two dimensions because the confederate applies force perturbations mostly from the vertical dimension. However, we agree that a bivariate analysis conducted by lumping both the variables would be more suitable considering the horizontal and vertical dimensions of the movements are not independent of each other. Hence, we conducted a mixed 2x2x10 ANOVA with Group (HV and LV) as between-subject factor and the Axis-dimension (horizontal and vertical) and Repetition (R1-R10) as the within-subject factors, for Experiment 1 and a mixed 2x2x5 ANOVA with Group (HV and LV) as between-subject factor and the Axis-dimension (horizontal and vertical) and Repetition (R1-R5) as the within-subject factors, for Experiment 2 and 3. The new analysis has been added to the revised manuscript. The results have been discussed in detail under Reviewer 2, pt. 8. Report of the analysis reads as following in the results section of Experiment 1, 2 and 3: 

Experiment 1 (page 19): “The mixed ANOVA revealed a main effect of repetition with Greenhouse-Geisser correction (ε =0.604), indicating that participants in both groups reduced their variability over time (F (9, 306) = 13.014, p<0.0001, η2 = 0.277, see Fig 5A). Post-hoc analysis revealed that spatial error at first repetition, R1 (mean=1.308, SE= 0.078) is significantly higher compared to other repetitions (all ps<0.005). The main effect of axis-dimension was also significant (F(1,34)=100.787, p<0.0001, η2 =0.748) with the variability on the horizontal dimension being larger (mean= 1.048, SE= 0.051) than the variability on the vertical dimension (mean= 0.769, SE= 0.042). The analysis also showed a significant main effect of group (F(1,34)= 7.249, p=0.011, η2 = 0.176), indicating that the HV group participants had a higher movement variability (mean= 1.029, SE=0.063) compared to the LV group (mean= 0.789, SE=0.063). The interaction between repetition and group (F (9,306)= 2.546, p= 0.008, η2 =0.070) was significant with Greenhouse-Geisser correction (ε =0.537), which was mainly driven by the higher variability of HV group participants at R1 (mean=1.549 , SE=0.111) compared to LV group (mean= 1.068, SE= 0.111). This was confirmed by post-hoc analysis (p=0.004). This indicates a faster reduction of variability by HV group compared to LV group. All other interactions were non-significant (ps >0.06; see Supporting Information S1 File).”

Experiment 2 (pg.29): “The mixed ANOVA revealed a main effect of repetition with Greenhouse-Geisser correction (ε =0.750), indicating that participants in both groups reduced their variability over time (F(4,152)=19.352, p<0.0001, η2 = 0.337, see Fig 7A). Post-hoc analysis revealed that spatial error at first repetition, R1 (mean=1.328, SE= 0.124) is significantly higher compared to other repetitions (all ps<0.0001) and R2 and R3 was also significantly different from R5 (ps<0.004). The main effect of axis-dimension was also significant (F(1,38)=63.396, p<0.0001, η2 = 0.625) with the variability on the horizontal dimension being larger (mean= 1.026, SE=0.088) than the variability on the vertical dimension (mean= 0.743, SE=0.062). The main effect of group did not reach a significance (F(1,38)= 0.095, p= 0.760, η2 =0.002) indicating that the variability of HV group (mean=0.862, SE= 0.104 ) participants was similar to LV group (mean=0.907, SE= 0.104). The interaction between repetition and axis-dimension (F(4,152)=8.716, p<0.0001, η2 = 0.187) was found to be significant with a Greenhouse-Geisser correction (ε =0.727). Interestingly, post-hoc revealed that the variability on the horizontal dimension was significantly higher than the vertical dimension in all 5 repetitions (all ps<0.004). All other interactions were non-significant (ps > 0.78; see Supporting Information S1 File).”

Experiment 3 (page 36): “The mixed ANOVA revealed a main effect of repetition, with Greenhouse-Geisser correction (ε =0.819), indicating that participants in both groups reduced their variability over time (F(4,148)=8.792, p<0.0001, η2 = 0.192, see Fig 8A). Post-hoc analysis revealed that spatial error at first repetition, R1 (mean=1.098, SE= 0.073) is significantly higher compared to other repetitions (all ps<0.05). The main effect of axis-dimension was also significant (F(1,37)=50.729, p<0.0001, η2 =0.578) with the variability on the horizontal dimension being larger (mean= 0.959, SE= 0.058) than the variability on the vertical dimension (mean= 0.737, SE= 0.047). The main effect of group did not reach a significance (F(1,37) = 0.666, p= 0.420, η2 =0.018) indicating that the variability of HV group (mean=0.889, SE= 0.072) participants was similar to LV group (mean=0.807, SE= 0.070). The interaction between repetition and axis-dimension was found to be significant (F(4,148)=3.118, p=0.037, η2 =0.078), with Greenhouse-Geisser correction (ε =0.642). Interestingly, post-hoc revealed that the variability on the horizontal dimension was significantly higher than the vertical dimension in all 5 repetitions (all ps<0.025). All other interactions were non-significant (ps >0.78; see Supporting Information S1 File).”

3.Ln 248 says that the task required “synchronous” matching with the partner– yet no information is provided about the temporal accuracy with learning. Relatedly the movement times also need to be provided to make sure that interpretation of changes in accuracy are not confounded by speed-accuracy trade-offs.

Reply: We agree that reporting synchrony measures is important for the interpretation of the data. Therefore, we have added to manuscript an analysis of the absolute asynchrony that assesses the synchronicity between the actors. Overall, the pattern of results indicates that asynchronies between the actors in HV groups are worse in Experiment 1 and 2, whereas it remained the same as LV group in Experiment 3. This suggests that the joint outcome was benefitted only when the partner was partially predictable. It could be that complete unpredictability or predictability of the partners highly variable movements hurts joint action. This is so because, in the former, there is no room for improving joint performance whatsoever, as partner is exerting force perturbations in random directions. In the latter, as the partner is completely predictable, predictability will be adopted as a key strategy to optimize the joint outcome. In this interpretation, a high variability of partners movements will only hurt the joint outcome. Hence, in Experiment 2, LV group reaps the benefit of having a predictable partner. In Experiment 3, since the partner is only partially predictable, there is room for improving the joint outcome. These results are also discussed in relation to individual performances under Reviewer 2 pt.8. 

 Regarding the comment about a potential speed-accuracy trade off in our study, we would like to point out that the inter-target interval in our design was cued and controlled for by the experimental script. Hence, the speed-accuracy trade-off cannot be a confounding factor that drives the changes in our accuracy results. Moreover, this left no room for systematic difference in movement time across conditions. 

The analyses of absolute asynchronies have been added in the revised manuscript. It reads as follows: 

Experiment 1 (pg. 21 ): “The ANOVA with a Greenhouse-Geisser correction (ε =0.729) on the absolute asynchronies between the two actors revealed a main effect of Repetition (F (9, 306) = 5.959, p<0.0001, η2 =0.149, see Fig 5D). The post-hoc revealed that the effect was mediated mainly by R1 (mean= 0.089, SE=0.003 ) being significantly lower to R2 (mean=0.100 , SE=0.002), R9 (mean=0.098, SE=0.003) and R10 (mean=0.103, SE=0.003) (all ps<0.01). The main effect of mean asynchronies between HV group (mean= 0.100, SE= 0.003) and LV group (mean= 0.092, SE= 0.003) showed a trend towards significance (F (1,34) = 4.117, p= 0.0503, η2 =0.108). The interaction between the two factors was not significant (F (9,306) = 0.701, p=0.708, η2 =0.020).”

Experiment 2 (pg.32): “The ANOVA conducted on the absolute asynchronies between the two actors revealed a main effect of Repetition (F (4, 152) = 2.439, p=0.049, η2 = 0.06, see Fig 7D). The post-hoc revealed that R1 (mean= 0.101, SE= 0.002) was significantly lower than R2 (mean= 0.107, SE=0.002) and R5 (mean= 0.106, SE= 0.002) (all ps<0.03), indicating that asynchrony increased over repetitions. The mean asynchrony between the two groups were also found to be significantly different (F(1,38)= 12.541, p= 0.001, η2 =0.248) with HV group having a higher asynchrony (mean= 0.107, SE= 0.001) compared to the LV group (mean= 0.101, SE= 0.001). The interaction between the factors was not significant (F (4,152) = 0.714, p= 0.583, η2 = 0.018).”

Experiment 3 (pg.40): “The ANOVA conducted on the absolute asynchronies of the two actors did not show a significant main effect of repetition (F(4,148)= 0.563, p= 0.690, η2 =0.015) indicating that asynchrony between the actors remained the same across repetitions (see Fig 8D). The mean asynchrony between the two groups (mean (HV) =0.102, SE= 0.002; mean(LV)= 0.097, SE=0.002) also did not reach significance (F(1,37)= 2.406, p=0.129, η2 =0.0.061), neither was the interaction between the factors significant (F (4,148) = 0.725, p= 0.576, η2 = 0.019).”

4.Some measure of performance on the ‘confederate’s side would also be critical to understand the performance reported here – since this is a person and not a robot, were there adjustments that the confederate made over time, or in different conditions that potentially affected the participant’s learning?

Reply: We agree that analysing the confederate’s data is needed for an adequate interpretation of the data. The analyses overall show a learning effect, as the confederate also improves her performance between first and last repetitions in all three experiments. On the one hand, this is a desired pattern as this is a natural pattern for the motor interaction between co-actors. On the other hand, this could be a confound with respect to the source of participants' learning. In fact, participants could be simply mimicking the confederate's movements. However, this is very unlikely, as participants and confederate never perform the same movements and never cover the same distance. Hence, participants would not improve their performance if they were only mimicking the confederate's movements.

 Unless the reviewer thinks otherwise, we will be adding the analysis to the supplementary material and not in the main manuscript as we believe the confederate’s learning performance is not a central part of our discussion. The analysis report reads as follows: 

Experiment 1 (S2 File): “The ANOVA on the confederate’s data with a Greenhouse-Geisser correction (ε =605) revealed a main effect of the repetitions, indicating that confederates in both groups showed reduction of spatial error over time (F (9,306) = 12.169, p<0.0001, η2 = 0.264). The analysis also showed a significant main effect of group, with the HV group (Mean= 1.377, SE = 0.092) having significantly larger errors than the LV group (Mean = 1.032, SE = 0.092), (F (1, 34) =7.075, p=0.012, η2 = 0.172). The interaction between the two factors was also significant (F (9, 306) = 7.646, p< 0.0001, η2 = 0.184), where R4, R5, R6 and R7 of HV group was significantly different from all other levels (all ps <0.05).”

Experiment 2 (S2 File): “The ANOVA with a Greenhouse-Geisser correction (ε =0.364) revealed a main effect of Repetition, indicating that confederates in both groups reduced their spatial error over time (F (4, 152) = 41.260, p< 0.0001, η2 = 0.521). However, spatial error of the HV group (Mean= 0.625, SE= 0.028) did not differ from that of the LV (Mean= 0.550, SE= 0.028) group (F (1, 38)= 3.578, p = 0.066, η2 = 0.086). The interaction between the factors was significant (F (4,152) = 4.686, p= 0.022, η2 = 0.110). Post-hoc analysis revealed that the interaction was driven by R1 and R2 of HV group which was significantly different from that of the LV group.”

Experiment 3 (S2 File): “The ANOVA with a Greenhouse-Geisser correction (ε =0.673) revealed a main effect of Repetition, confirming that confederates in both groups learned to reduce their spatial error over time (F (4, 148) = 6.010, p< 0.0001, η2= 0.140). There was no main effect of Group (F (1, 37) = 2.90, p = 0.097, η2 = 0.073), indicating that spatial error did not differ significantly between HV group (Mean= 1.152, SE= 0.121) and LV group (Mean= 0.864, SE= 0.118). The interaction between the factors was significant (F (4,148) = 7.235, p <0.001, η2 = 0.164). The post-hoc revealed that R3 of HV group was significantly different from the other levels (p=0.005).”

5.Finally, I felt that the authors’ underlying theoretical motivation needs to be explained better in terms of the task used here. The authors refer to Wu et al and exploration, but the tasks in Wu et al (shape matching and force field adaptation) *require* exploration to move from one movement pattern to another. Here in the sequence learning paradigm, it is not clear *why* exploration is required to perform this task (and therefore why variability should be useful). Instead, the changes reported in accuracy/precision are simply refinements of an existing movement pattern. The authors need to address this more carefully both in the Intro and the Discussion and mention why exploration is required for learning this sequence task.

Reply: In general, action exploration is beneficial for motor learning as it allows exploration of novel actions required to constantly update the internal model of the action. This would benefit performance as it ensures that the agent can adapt to changing circumstances in her action space. Variability proves to facilitate action exploration by allowing agents to alter their motor parameters- from gross to fine-grained motor refinements of an existing movement pattern. The Wu et al., results demonstrate a general learning principle where increased variability enables faster learning. Our task not only requires participants to aim and hit at specific target locations, but also to learn to adapt to the various force perturbations that they are exposed to. Participants at our task are not dealing with discrete action responses, rather with movement trajectories from one target to another, during which participants experience constant force perturbations. Hence, we would like to argue that our task does require action exploration just as in the force field adaptation task in Wu et al., studies. Our paper explores, in what specific joint action context does the partner’s variability foster action exploration in individuals. We have elaborated this point and added it to the intro and discussion part of the revised manuscript. The edited part can be seen in our reply to Reviewer 2 pt.8 (under ‘Minor:’).

Minor:

1. Ln 28 – “individual motor learning studies demonstrate individual’s ability//”. The term “individual studies” is confusing. 

Reply: We have revised this to “Motor learning studies demonstrate that an individual’s natural motor variability predicts the rate at which she learns a motor task.” (pg. 2) 

2. Ln 58 “in such joint actions and in joint actions” – rephrase

Reply: This has been rephrased as “Generally, in such joint actions, individuals adopt various coordination strategies to minimize error and improve joint performance” (pg. 3) 

3.Ln 65 – “making one’s own movement less variable and more consistent”. Is less variable the same as more consistent or does it refer to some other feature here?

Reply: This phrasing was confusing and has been deleted during revision. We meant making one’s movement less variable in time. 

4.Ln 76 – the paragraph on interpersonal coordination seems a little out of place here. Either develop this more fully or move it to the Discussion if it is nots central to the introduction

Reply: We agree and have removed this paragraph from the text. 

5.Ln 95-98: The study referred here actually showed that the magnitude of variability was not important – rather it was the structure of the variability (as measured by the autocorrelation) that determined learning rate.

Reply: Correct. The text has been edited to clarify this point in the manuscript. It now reads as follows (pg. 6): “Similarly, Barbado and colleagues [14] showed that the magnitude of variability in error-based learning only negatively influences the learning rate, while the structure of individuals’ variability or the systematic variation in their movements enhanced their ability to detect motor error, which ultimately led to faster learning.”

6.Ln 103-104: The two statements here referring to [19,20] need to be explained in a bit more detail to help understand why variability may play multiple roles.

Reply: The text has been edited as follows (pg. 5) : “Ranganathan and colleagues [19], [20] propose that even though variability can enhance action exploration in certain types of learning, a higher magnitude of variability may adversely affect learning by leading to poor retention of learned solutions. Also, the learning will be adversely affected if the mechanism involved is use-dependant, i.e., if the learning requires one to produce subsequent movements similar to the previous ones or if it requires coordination pattern stability as practising unstable movement patterns can lead to poor learning.”

7.Ln 118-130 seems to be talking about generalization – yet this is not a major focus of the paper.

Reply: These lines have been removed from the manuscript. 

8.Ln 139-142. The two options spelled out here are interesting, but do not seem to be tested in the paper at all.

Reply: With the new analysis on the joint performance that has been included in the revised manuscript, we managed to pull out discussion about the strategy adopted by individuals to regulate their own and their partner’s variability for optimizing individual learning and joint outcome. We have revised the general discussion of the manuscript accordingly. It now reads as follows (pg. 44): 

“During joint action coordination that involves physical coupling, motor variability produced by an actor can have a direct impact on the partner’s movements. In the current study, we investigated whether individuals could regulate their own and their partner’s variability when learning a motor task together. Specifically, we looked at how the variability at different degrees of predictability of force perturbations coming from a partner, during a joint action, foster or hamper individual and joint performance and what strategies unfolds in this context, so that individuals can optimize both individual and joint outcome. Motor learning generally involves reduction of variability or error in one’s movement over practice. Action exploration and subsequent exploitation of the best explored actions are found to be aiding this learning process in specific types of learning. In our study, participants could either utilize their own variability, their partner’s variability, or both, to explore possible motor refinements that allow successful learning.

 The finding on participants’ movement variability showed that in all three experiments, participants were generally more variable on the dimension that was less constrained by haptic coupling, that is the horizontal dimension. Also, participants learned to reduce their movement variability and, consequentially, their spatial error over time in all three experiments. This indicates that individuals were able to utilize their internal variability to explore their motor parameters relevant for the task which benefitted their individual performance. However, individuals might adopt different strategies when it comes to utilizing external variability coming from their partner, during a joint action. 

 In Experiment 1, where partner’s variability was unpredictable, it was observed that individuals learned to reduce their spatial error over time, but HV group participants were worse than LV group in their performance. Also, the correlational analysis showed that when the partner’s movements were not predictable, individuals were unable to compensate for large perturbations. These results suggest that the high variability of the partner was not providing any relevant information for the task as participants’ movements were perturbed in random directions. In other words, partner’s high and unpredictable variability was detrimental for individual performance. However, the fact that participants still learned to reduce their spatial error over time points to the possibility that they modulate their own variability to ensure exploration and improvement in performance. In Experiment 2 and 3, when the partner’s movements were completely or at least partially predictable, spatial accuracy was not impacted. Importantly, it was observed that individuals compensated equally well for a wide range of force perturbations when the partner’s movements were at least partially predictable, implying resilience to larger perturbations. These results suggest that predictability becomes a necessary pre-condition for partner’s variability to positively influence individual’s performance. It was also observed that participants were able to offset the group difference in Experiment 2 and 3 already at the first repetition, while this was not the case in Experiment 1. This shows that participants could utilize the knowledge about the target locations they gained from the individual phase, whereas in Experiment 1, this knowledge was not enough to offset the difference between groups. This also indicates that predictability of partner’s movements is crucial in this kind of joint action. Given that participants’ regulated their own movement variability, as observed in the variability analysis of Experiment 2 and 3, it is possible that they were relying both on their own and their partner’s variability for the benefit of their individual performance. Taken together the results on individual performances of the three experiments, it seems that the magnitude of variability does not entail benefit of learning, rather the structure of task-relevant variability coming from the partner, is what seems to enhance action exploration. 

 On a mechanistic level, we propose that under conditions of predictability, a varied range of force perturbations experienced from partner allows individuals to generate a flexible internal model accommodating various movement solutions for a wider range of task parameters (in our case, various force perturbations) [21]. Thus, in the high-variability condition, instead of forming a single motor plan required to perform the task under a constant force as in the low-variability condition, individuals generated multiple motor plans for performing the task under varying task environment. Depending on the perturbation experienced, the most optimal plan could be exploited for improving the efficiency of outcome achievement. This allowed them to exhibit similar performance as in the low-variability condition (Experiment 2 and 3). Had a wider action exploration not been fostered by partner’s variability in these cases, participants in the high-variability group should have performed worse than the low variability group due to the more varied force perturbations.

 It is important to note that the individual performance does not necessarily reflect the success of joint outcome. The results of the analyses on joint performance show that participants in the HV group in Experiment 1 and 2 were particularly worse than participants in the LV group at synchronising with their partners (significant main effect of group). There was also a trade-off between the spatial accuracy and the temporal coordination between the actors in Experiment 1 and 2. It was shown that the asynchrony increased while the spatial accuracy was decreasing over repetition. This was not the case in Experiment 3. This means that the high variability coming from a partner, even when predictable, was not beneficial for the joint outcome. On the other hand, the results of individual movement performance show a different pattern, as mentioned before. While the partner’s high variability seemed to hamper performance in Experiment 1, the partners high variability produced in a predictable manner in Experiment 2, allowed individuals to explore their action space and gather motor information relevant for the task. This was reflected in a benefit of individual performance. However, coordinating with a partner that is completely predictable also means that there is no room for further exploration to improve joint performance. This means, in Experiment 2, conditions for optimal joint performance are hard-wired, i.e., predictability of partner’s movements can be adopted as a coordination strategy for improving the joint outcome. Therefore, because predictability of the partners movements anchors the joint performance, high variability cannot be integrated in movement exploration and hence, it hurts the joint outcome. Interestingly, in Experiment 3, when the variable forces were introduced in an only partially predictable manner, actors maximally exploited action exploration to benefit their joint outcome even when variability coming from the partner was high. The same pattern was observed in their individual performance. This shows that partial predictability of partners movements allowed individuals to benefit both in their individual and joint performance. It could be that partial predictability provides more room for motor adjustments, unlike in the complete predictability condition, which in turn aids optimization of the joint outcome. In other words, highly variable partners, while being only partially predictable, may promote individuals to learn and thus improve joint performance as well as individual performance. Similar findings were demonstrated in joint action scenarios were actors improved their individual performance over time while achieving high synchronization in a joint tracing task when the task conditions provided room for mutual predictions and adaptation to each others’ movements [22]. 

Our findings indicate that individuals might switch their action exploration strategy to optimize individual performance, joint performance or both depending upon the structure of variability provided to them by their partner. It is possible that individuals might selectively rely on either their own or on the partner’s variability for benefitting individual, joint or both outcomes, depending on the joint action contexts in which the interaction takes place. However, it remains to be seen how the individual and joint performances will differ if both actors hold symmetrical knowledge about the task. This is an interesting question to explore in future research.”

9.Ln 230 – remove the “/sec” - just 120 Hz

Reply: Corrected. 

10.In all Experiments, was any feedback provided to the participants (in terms of errors or times etc.)?

Reply: No, there was no feedback in terms of their accuracy. Participants were only given a cue for a temporal window within which they had to move from one target to another (inter-target interval).

11.Were data from the “individual action phase” analyzed at all – it seems like this would be important to understand how the learner changed strategies in the joint learning task.

Reply: Please see also our reply to Reviewer’s 1 point 2. We did not analyse individual-phase data, as this phase was only for task familiarization. The individual phase does not reflect the actual task as it does not involve any force perturbation that participants must learn to adapt. Rather, the individual phase only familiarizes the participants with the target position of the sequences they must learn to perform with perturbations from a partner during the joint phase. Thus, comparisons between participant’s performance in the individual and joint phase are not interpretable in the current study.

12.Ln 345 – please report exact p-values unless they are very small (Say <.001). Same comment holds for other part of the manuscript as well.

Reply: The manuscript has been edited accordingly.

13.Ln 358-365: The number of data points on Fig 5B seem larger than the total number of participants in the HV group.

Reply: This is because each data point depicts the average spatial error produced by all participants in the HV group while a force on the y-axis was produced. Thus, the number of data points reflects the number of different distances between the two target points in a target configuration roughly corresponding to the different levels of force that could occur in this experiment.

However, we would like to bring to your attention that there was a mistake in the values reported in the correlation analysis in the original manuscript. We apologize for the mistake and would like to include the correct analysis in the edited manuscript. The correct analysis does not change any of the current results, however the values are slightly different from the originally reported analysis. The corrected report has been added to the manuscript. The edited texts read as follows: 

Experiment 1 (pg. 21): “There was a positive correlation between the force perturbation (Mean Force= 20.253, SD= 6.817) and spatial error (Mean Error=1.034, SD= 0.304) which was statistically significant (r = 0.728, n= 32, p< 0.0001).”

Experiment 2 (pg. 31): “There was no significant correlation (r= -0.392, p= 0.064, n= 23) between force perturbation (Mean Force= 20.734, SD= 6.036) and spatial error (Mean Error= 0.784, SD= 0.185, see Supporting Information, S2 File for r values at different repetitions, separately).”

Experiment 3 (pg. 40): “There was no significant correlation between force perturbation and spatial error (r = 0.264, n= 31, p=0.150, see Supporting Information, S2 File for r values at different repetitions, separately).”

14.Ln 365 – Not sure what this means. The author say that the LV group experienced variation (just lower in magnitude) in Ln 257– so why is the LV not plotted here?

Reply: LV group sequences were built by combining target configurations that produced force perturbations with a very small variance between them. This means that the number of different force levels in the LV group is not sufficient to perform a correlational analysis, contrary to the wide range of forces in the HV group. The line has been rephrased for better clarity and it now reads as follows (pg. 21): 

“As the LV group experienced a significantly lower range of varying forces leading to an insufficient number of force magnitudes to perform a similar analysis, the correlation was not done on the LV group in the current and following experiments.” 

15.Ln 363 – not clear what the authors mean by “correlation is significant even without removing the outliers”. If they are truly outliers, they need to be removed because they will affect the magnitude of the correlation (even if the significance is not affected) 

Reply: We agree. The revised version of manuscript includes only the analysis post- outlier removal (mean (+/-) 2.5*SD).

16.Ln 455 – please use full sentences to describe this.

Reply: Done.

17.At several places, the authors seem to go back and forth between the terms ‘Variability’ and “Group’ to describe the between subject factor – in my view ‘Group’ is much clearer since there are only two groups.

Reply: We have replaced ‘Variability’ with ‘Group’ throughout the manuscript.

---

## [Decision Letter · Decision Letter 1]

26 Aug 2020

PONE-D-20-03780R1

How does a partner’s motor variability affect joint action?

PLOS ONE

Dear Dr. Sabu,

Thank you for submitting your manuscript to PLOS ONE. After careful consideration, we feel that it has merit but does not fully meet PLOS ONE’s publication criteria as it currently stands. Therefore, we invite you to submit a revised version of the manuscript that addresses the points raised during the review process. Reviewer 2 still has some concerns regarding interpretation - I'm not going to reject a manuscript on impact, but do seek assurances that the data represent what they are purported to represent, with all suitable caveats made where appropriate for more ambiguous findings.

We look forward to receiving your revised manuscript.

Kind regards,

Gavin Buckingham

Academic Editor

PLOS ONE

Reviewers' comments:

Reviewer's Responses to Questions

**Comments to the Author**

1. If the authors have adequately addressed your comments raised in a previous round of review and you feel that this manuscript is now acceptable for publication, you may indicate that here to bypass the “Comments to the Author” section, enter your conflict of interest statement in the “Confidential to Editor” section, and submit your "Accept" recommendation.

Reviewer #1: All comments have been addressed

Reviewer #2: (No Response)

2. Is the manuscript technically sound, and do the data support the conclusions?

Reviewer #1: (No Response)

Reviewer #2: Partly

3. Has the statistical analysis been performed appropriately and rigorously? 

Reviewer #1: (No Response)

Reviewer #2: No

4. Have the authors made all data underlying the findings in their manuscript fully available?

Reviewer #1: (No Response)

Reviewer #2: Yes

5. Is the manuscript presented in an intelligible fashion and written in standard English?

Reviewer #1: (No Response)

Reviewer #2: Yes

6. Review Comments to the Author

Reviewer #1: (No Response)

Reviewer #2: I have reviewed the revised manuscript submitted by Sabu et al. I thank the authors for the clarifications and the extensive changes but I feel that there are still 2 main concerns:

1. My concern about the predictability argument does not seem to be resolved. The authors show analysis of the 'first block' (the forces in the first block *has* to be 'unpredictable' by definition because there is no other prior experience from which the participant can predict anything). Yet the differences exist between Exp1 and 2/3 even in this block. I think this *severely* undermines the argument about predictability of forces being the main driving factor.

In their response, the authors argue "We would like to argue that this could be because the pre-exposure to the target locations in the individual phase was enough for the participants to offset the group effect while learning the task together with a partner producing atleast partially predictable movements". I'm not entirely sure what this means. How could the exposure to the target locations in the individual phase (when there were no forces involved) help the learner 'predict' the forces produced by the confederate in Exp 2/3 the first time they experienced these forces? To me, this indicates that some other confound is present in the data

Relatedly, the finding that the confederate also showed 'improvements in performance' and differences between conditions is worrying because it creates a confound in interpretation of the learner's performance. Because the confederate is haptically coupled to the learner the confederate's errors and the related forces they exert also influence the learner's spatial error. (i.e. imagine if the learner had the intent to make the exact same motion each time - the movement of the confederate would influence how the spatial error of the learner). This is also why I disagree with the characterization that spatial error is an 'individual' performance metric in Ln 172 - when the forces are coupled spatial error is no longer an individual measure of performance.

2. Moreover, related to my concern about theoretical rationale, after reading the response from the authors which provided clarification, I am still unsure how this paper has anything to say about 'learning' in the traditional sense of the word (see performance-learning distinction - see for e.g. Salmoni, Schmidt & Walter, 1984). What the paper measures is 'performance' and its change over time. However, since there is no 'common condition' under which the groups are tested (e.g., a test where all groups switched to the same haptic condition), I believe no inferences about learning (which reflects the underlying change in motor skill) can be made from the data. So it is still not clear to me that the the framing of the intro and discussion in terms of Wu et al. and learning/exploration is appropriate here (I think the framing in terms of joint action and how someone regulates their variability in response to a partner's variability is OK)

Minor:

1. It is not specified what the movement times are for these movements (only the cueing time of 1000ms is mentioned). Also I was referring to the speed-accuracy tradeoff for movement time (i.e. Fitts' Law). Movement times here are controlled by the participant and not by the script.

2. Ln 332-333: I think the outlier stats need to go under Ln 345 where the outlier criteria is mentioned

3. Ln 801 - if the authors wish to compare the magnitude of the horizontal to the vertical variability, then it should be normalized to the amplitude of the motion (i.e. a higher amplitude motion along the horizontal will give rise to higher variability along that dimension because of signal dep noise)

4. Ln 536 - I'm not sure if this is what the authors intended but the spatial error (what I Had referred to as bivariate error) is not the average of the x- and y- components (since it also has take into account the correlation between them in diagonal movements for example).See for example (Hancock, Butler, and Fischman (1995))

5. For the Bayesian test, please indicate what prior was used. Also in Table 1, the N column needs to be made wider so that 20 is in the same row

6. For the correlations, there are both within- and between-subject data pooled to make a correlation- this inference can be errorneous (see for eg. Simpson's paradox). Either this correlation needs to be done at an individual level and then averaged or the nested nature of this data needs to be accounted for. Also this is a correlation between two continuous variables - so I don't fully understand why the LV group cannot be plotted (yes, their range will be smaller and this can be acknoweldged in the Discussion, but it would be helpful to see the correlation plots anyway)

7. PLOS authors have the option to publish the peer review history of their article (what does this mean?). If published, this will include your full peer review and any attached files.

Reviewer #1: No

Reviewer #2: No

---

## [Author Response · Author response to Decision Letter 1]

10 Oct 2020

Responses are attached as a separate document under the title 'Response to Reviewers'.

---

## [Editor Report · Decision Letter 2]

15 Oct 2020

How does a partner’s motor variability affect joint action?

PONE-D-20-03780R2

Dear Dr. Sabu,

We’re pleased to inform you that your manuscript has been judged scientifically suitable for publication and will be formally accepted for publication once it meets all outstanding technical requirements.

Kind regards,

Gavin Buckingham

Academic Editor

PLOS ONE
---

## [Editor Report · Acceptance letter]

20 Oct 2020

PONE-D-20-03780R2 

How does a partner’s motor variability affect joint action? 

Dear Dr. Sabu:

I'm pleased to inform you that your manuscript has been deemed suitable for publication in PLOS ONE. Congratulations! Your manuscript is now with our production department. 

Kind regards, 

on behalf of

Dr Gavin Buckingham 

Academic Editor

PLOS ONE